

# Improving Forecasts of Persistent Contrails through Ice Deposition Adjustments

Zane Dedekind[1], Alexei Korolev[1], and Jason A. Milbrandt[1]

[1]Meteorological Research Division, Environment and Climate Change Canada, Toronto, Ontario, Canada

**Correspondence:** Zane Dedekind (zane.dedekind@ec.gc.ca)

**Abstract.** Aviation-induced clouds, especially persistent contrails, contribute significantly to anthropogenic climate forcing, often surpassing the short-term impact of aviation $CO_2$ emissions. These clouds form in ice-supersaturated regions, where they trap longwave radiation and warm the climate. On 25 November 2023, widespread ice-supersaturated layers over eastern Canada and the USA led to extensive contrail formation, confirmed by GOES-16 satellite imagery and ground-based pho-
tography. Atmospheric conditions were characterized using ceilometer data from Toronto Pearson Airport and radiosonde soundings.

High-resolution simulations were conducted using the Global Environmental Multiscale (GEM) model with the Predicted Particle Properties (P3) microphysics scheme. The Contrail Avoidance Tool (CoAT), incorporating Schmidt-Appleman Criteria and a wake vortex model, simulated persistent contrail formation and properties. Sensitivity tests adjusting ice depositional
growth rates evaluated their impact on ice supersaturation. Results indicate that the control (CNTL) simulation underestimated relative humidity over ice ($RH_i$), a common limitation where moisture is depleted too rapidly. Reduced depositional growth rates improved $RH_i$ forecasts and contrail-forming regions. However, GEM-CoAT underestimated contrail depth and ice number concentration in very shallow high-$RH_i$ layers. CoAT simulations also revealed that SAC alone is insufficient, as wake vortex dynamics can induce adiabatic warming, leading to ice particle sublimation.
Further analysis examined contrail formation for two aircraft types (A321 and B747). The B747 generated deeper wake vortices, enhancing adiabatic heating and reducing contrail ice number concentrations by 27% in sensitivity simulations and 78% in the CNTL simulations. Adjusting depositional growth rates allowed GEM-CoAT to accurately simulate contrail formation and persistence.

## 1 Introduction

Contrails, or condensation trails, are linear ice clouds that form in the wake of aircraft flying at high altitudes where the ambient atmosphere is sufficiently cold, typically colder than $-40\,°C$, and humid. These artificial clouds originate from the mixing of hot, moist exhaust gases with the surrounding air, leading to rapid condensation and freezing of water vapor onto emitted



particles, such as soot and other aerosols (Schumann, 1996; Kärcher, 2018). While some contrails dissipate quickly, others
persist and spread into contrail cirrus if the environmental air is ice-supersaturated, which can contribute to cloud cover and
significantly alter the radiative balance of the Earth's atmosphere (Kärcher, 2018; Lee et al., 2023).

The aviation industry is a significant contributor to anthropogenic climate change, accounting for approximately 3.5 %
of total effective radiative forcing (ERF) as of 2018 (Kärcher, 2018; Lee et al., 2021). Notably, about two-thirds (66 %) of
this warming impact is attributed to non-$CO_2$ effects, primarily from contrail cirrus and nitrogen oxide ($NO_x$) emissions. The
radiative forcing from contrail cirrus alone is estimated to be $57.4\,\mathrm{mW\,m^{-2}}$, making it the single largest contributor to aviation-
induced climate effects, surpassing the impact of cumulative $CO_2$ emissions from aviation, which is $34.3\,\mathrm{mW\,m^{-2}}$ (Lee et al.,
2021).

Contrail cirrus exerts a complex influence on Earth's energy balance. While they reflect some incoming solar radiation, their
primary effect is trapping outgoing longwave radiation, leading to a net warming impact. The strength of this effect depends
on atmospheric conditions, contrail properties, and flight patterns. Studies indicate that contrail-induced cloudiness covers up
to 10 % of the sky area in high-traffic regions such as Europe and North America (Burkhardt and Kärcher, 2011).

Given the projected growth in global aviation, expected to double its $CO_2$ emissions by 2050 without mitigation strategies
(Lee et al., 2009), understanding and addressing contrail-induced climate effects is a critical research priority. Several mitiga-
tion strategies have been explored, including optimizing flight altitudes and routes, reducing soot emissions by transitioning to
low-aromatic or biofuel blends, using hydrogen fuels, and employing alternative aircraft propulsion technologies (Burkhardt
et al., 2018; Teoh et al., 2020, 2022; Lottermoser and Unterstraßer, 2025). Studies suggest that targeted rerouting of just 2 %
of flights could reduce contrail radiative forcing by up to 59 % with negligible fuel penalties over Japanese airspace while
rerouting 12 % of flight reduce the radiative forcing by 80 % over the North Atlantic (Teoh et al., 2020, 2022).

Simulating the evolution of contrail ice number concentration (CINC) is crucial for accurately predicting contrail persis-
tence and lifetime. Ice crystal number concentration directly influences contrail optical depth, growth rates, and sedimentation,
which in turn determine their overall climate impact. Higher CINCs lead to greater optical thickness and longer persistence,
whereas lower CINCs promote faster ice particle sedimentation and sublimation, reducing contrail lifespan. Several parameter-
izations have been developed to better represent contrail microphysics and ice crystal evolution. Schumann (2012) introduced
the Contrail Cirrus Prediction Tool (CoCiP), a fast process-based model designed to estimate contrail properties based on
meteorological conditions and aircraft emissions. Lewellen and Lewellen (2001); Lewellen et al. (2014) applied large-eddy
simulations to investigate wake vortex dynamics and their impact on contrail ice crystal formation, showing how turbulence in-
fluences contrail spreading and persistence. Unterstrasser (2016) developed a parameterization that enhances the representation
of contrail-to-cirrus transitions, improving the understanding of how environmental conditions, including ice supersaturation,
affect contrail growth and dissipation.

The persistence and variability of ice supersaturation in ice clouds are strongly influenced by the deposition coefficient ($\alpha_D$),
which describes the efficiency of water vapor molecules attaching to ice crystal surfaces. Laboratory studies, including wind





tunnel experiments, cloud chambers, and cold-stage scanning electron microscopy imaging, have shown that $\alpha_D$ is not constant
but varies with temperature, supersaturation, and ice crystal habit. Measurements by Fukuta and Takahashi (1999), Lamb et al. (2023), and Harrington and Pokrifka (2024) reveal that $\alpha_D$ often falls well below unity under cold, upper-tropospheric conditions typical of cirrus, indicating kinetic limitations to depositional growth. These studies also identify supersaturation thresholds associated with morphological transitions such as hollowing, providing physical constraints on when and how efficiently crystals grow.

From a modeling perspective, simplified treatments that assume $\alpha_D = 1$ or apply saturation adjustment fail to capture the observed frequency and magnitude of ice-supersaturated regions. Gierens et al. (2003, 2020) and (Sperber and Gierens, 2023) showed that such assumptions result in systematic underprediction of supersaturation and contrail persistence in numerical weather and climate models. To address this, Kärcher et al. (2023) incorporated supersaturation-dependent deposition coefficients, based on laboratory constraints from Lamb et al. (2023), into stochastic parcel models coupled with gravity wave-induced temperature fluctuations. Their simulations reproduced both the mean and variability of in situ supersaturation measurements, highlighting the role of depositional kinetics in ice cloud microphysics. Together, laboratory and modeling work shows the need for physically-based representations of $\alpha_D$ to accurately simulate ice cloud evolution.

This study investigates how variations in the depositional growth of ice, calculated by adjusting $\alpha_D$, affect the extent and persistence of ice-supersaturated regions and the properties of contrails within a high-resolution numerical weather prediction framework. We explore the following questions:

– How does the adjustment of the ice deposition rate, through changes in the deposition coefficient, influence the persistence and intensity of ice-supersaturated regions?

– In what ways does the deposition rate impact contrail formation processes and the resulting CINC within young contrails?

– To what extent do wake vortex dynamics, particularly adiabatic heating during plume descent, limit contrail formation despite favorable Schmidt–Appleman Criteria (SAC) conditions?

The remainder of this paper is organized as follows: Section 2 describes the methods, including an overview of the synoptic environment, observational datasets, and the modeling tools used to simulate contrail formation. Section 3 presents the results of sensitivity experiments examining how variations in the depositional growth rate affect the $RH_i$, contrail ice number concentrations, and contrail persistence. Section 4 discusses the implications of these findings. The paper concludes with a summary of key findings and recommendations for improving contrail representation in weather and climate models. Appendix A provides the physical basis and equations governing the depositional growth of ice crystals and how the deposition coefficient $\alpha_D$ influences ice particle mass growth under supersaturated conditions.



## 2 Methods

### 2.1 Synoptic conditions and observational data

On 25 Nov 2023, a large ice-supersaturated region formed, leading to extensive cirrus cloud formation over the southeastern and northeastern parts of Canada and the USA, respectively. The upper-level atmospheric conditions included a deep trough in the jet stream which allowed cold Arctic air to plunge southward into the Great Lakes region. This created significant temperature contrasts and instability in the upper troposphere. Evidence from radiosonde soundings, satellite observations, and surface-based photography confirmed the presence of ice-supersaturated layers aloft, within which cirrus clouds and persistent contrails formed (Fig. 1b-d).

#### 2.1.1 Radiosonde data

Upper-air radiosonde data were obtained from the University of Wyoming dataset (University of Wyoming, 2024). The radiosonde data encompassed the balloon's precise trajectory, detailing latitude, longitude, altitude, and time. This information is crucial, as radiosondes can drift significantly from their launch sites during ascent. Balloons may drift approximately 5 km in the mid-troposphere, around 20 km in the upper troposphere (Seidel et al., 2011). Radiosonde data was collected to determine the drift distance between the launch location and the 300 hPa pressure level (approximately jet cruising altitude) for various stations at 1200 UTC on 25 November 2023 (Table 1).

**Table 1.** Radiosonde Drift Distances at 300 hPa at 1200 UTC on 25 Nov 2023

| Station Name | Location | Station Identifier | Drift Distance (km) |
|---|---|---|---|
| Albany | NY, USA | ALB | 31.23 |
| Gaylord | MI, USA | APX | 25.22 |
| Buffalo | NY, USA | BUF | 23.23 |
| White Lake | MI, USA | DTX | 19.96 |
| Green Bay | WI, USA | GRB | 23.44 |
| International Falls | MN, USA | INL | 34.12 |
| Maniwaki | QC, Canada | WMW | 42.42 |
| Pickle Lake | ON, Canada | WPL | 41.82 |

Assuming a balloon ascent rate of $5\,\mathrm{m\,s^{-1}}$, the time to reach the 300 hPa pressure level is approximately 30.5 min if the balloon drift is ignored. However, considering potential variations in ascent rates and balloon drift, the actual time can range between 40 min and 110 min. For modeling purposes, we assume an ascent time of approximately 50 min to align the model data with the balloon's arrival at the 300 hPa level.





### 2.1.2 Satellite data

The Advanced Baseline Imager (ABI) aboard the GOES-R series satellites is a passive imaging radiometer featuring 16 spectral bands, including 10 in the infrared spectrum. The spatial resolution for these infrared bands is 2 km (Kalluri et al., 2018). In its operational modes, the ABI provides full-disk imagery every 10 min, images of the contiguous United States (CONUS) every 5 min, and two mesoscale images every 60 s (or one every 30 s). Among its capabilities, the ABI utilizes a "Dust" RGB (Red-Green-Blue) composite to detect and monitor airborne dust. This product is also especially useful to detect contrails. This

composite combines data from infrared channels 8.4 $\mu$m (Band 11), 10.3 $\mu$m (Band 13) and 12.3 $\mu$m (Band 15). The GOES-16 data was downloaded from University of Utah (2020) and the "Dust" RBG was generated using the software package from Blaylock (2023).

### 2.1.3 Aircraft flight data

– **Data Source:** Flight data was obtained from Flightradar24 (Flightradar24, 2024).

– **Selected Flights:** A subset of flights was identified as potential contributors to contrail formation (Fig. 1 and Table 2). These flights cruised at altitudes ranging from 8,800 m to 10,700 m (29,000 ft to 35,000 ft).

– **Recorded Data Points:** Flight data was recorded at 30 s intervals, during which aircraft traveled approximately 6 to 7 km at cruising speed between recordings. The data was then interpolated onto a 1 km × 1 km grid, aligning with the model grid resolution, ensuring one recording per grid point.

**Table 2.** Aircraft Wingspan and Cruising Altitude Data

| Aircraft Identifier | Aircraft Type | Wingspan (m) | Cruising Altitude (ft) | ∼Time of Toronto Flyover (UTC) |
|---|---|---|---|---|
| **TK6061** | **B747** | **64.4** | **31,000** | **1345** |
| **CV6686** | **B747** | **64.4** | **34,000** | **1315** |
| AS459 | B747 | 64.4 | 34,000 | 1330 |
| **DL384** | **A321** | **35.8** | **30,000–34,000** | **1245** |
| AA2455 | A321 | 35.8 | 34,000 | 1330 |
| UA364 | A319 | 35.8 | 34,000 | 1415 |
| UA311 | B757 | 38.0 | 34,000 | 1345 |

### 2.1.4 Ceilometer at CYYZ

The Ceilometer, CHM 15k Cloud Height Meter, is an advanced light detection and ranging (LIDAR-based) remote sensing instrument deployed at Toronto Pearson International Airport (CYYZ) to measure cloud height, penetration depth, and vertical visibility. Operating at a 15 s temporal resolution, it employs the lidar technique to emit short laser pulses into the atmo-







**Figure 1. (a)** Interpolated flight paths and sounding stations **(b)** GOES-16 Dust RGB and sounding stations at 1430 UTC **(c)** and **(d)** photos taken from Toronto, Canada at 1420 UTC by Alexei Korolev.

sphere. These pulses scatter upon interaction with aerosols and cloud particles, with the backscattered signal being analyzed to

determine cloud structure and visibility conditions.





The CHM 15k is capable of measuring up to 15 km in altitude with a range resolution of 5–15 m, depending on the measurement mode. Its full waveform analysis allows for the identification of multiple cloud layers (up to 9), 3 layers in its current configuration, and provides high-resolution backscatter profiles. It utilizes photon counting technology, enhancing detection sensitivity and minimizing background noise. The system records parameters such as cloud base height, penetration depth, maximum detectable range, vertical visual range, and sky condition, making it crucial for aviation meteorology and atmospheric research.

## 2.2 Model

### 2.2.1 Atmospheric model and initialization

The Global Environmental Multiscale (GEM) model is a versatile atmospheric modeling system widely used for high-resolution simulations of atmospheric processes (Côté et al., 1998; Girard et al., 2014). The integration of the Predicted Particle Properties (P3) microphysics scheme into GEM represents a significant advancement in the modeling of ice-phase hydrometeors and mixed-phase cloud dynamics. This methodology outlines the configuration and application of the GEM-P3 setup as described in several studies (Morrison and Milbrandt, 2015; Milbrandt and Morrison, 2016; Milbrandt et al., 2021; Qu et al., 2022; Cholette et al., 2024; Korolev et al., 2024).

GEM has a non-hydrostatic, fully compressible dynamical core using semi-Lagrangian advection and a terrain-following hybrid vertical coordinate system (Côté et al., 1998; Girard et al., 2014). This configuration is suitable for a wide range of resolutions, from mesoscale to cloud-resolving scales. Our simulations use a horizontal grid spacings of $1\,km \times 1\,km$ over a $1600 \times 1000$ domain centered at $45.2\,°N$ and $-79.9°E$ (eastern Canada). Typically, 60 vertical levels are used in operational systems using GEM, but here 121 vertical levels are used to enhance the vertical resolution in the upper tropical troposphere with grid spacings of $\sim 230\,m$ at $300\,hPa$ to capture ice supersaturated regions better. GEM uses lateral and surface boundary conditions obtained from the operational 2.5-km High Resolution Deterministic Prediction System (HRDPS; Milbrandt et al., 2016), which is updated hourly. A 30 s model timestep is used, with the model initialized at 0600 UTC and running for a duration of 13 h.

### 2.2.2 Cloud microphysics scheme

The P3 microphysics scheme is unique in that the ice phase uses the property-based approach, in contrast with the traditional approach of predefined ice-phase categories, whereby all ice-phase or mixed-phase particles are represented by one or more generic or "free" categories whose bulk physical properties evolve freely and continuously. This enables each ice category to represent a wide range of dominant types of ice. With the use of multiple free ice-phase categories, multiple modes – i.e. populations of ice with different bulk properties – can exist in the same point in time and space. A complete description of the original P3 scheme can be found in Morrison and Milbrandt (2015) and Milbrandt et al. (2016); descriptions of subsequent





major developments can be found in Milbrandt et al. (2021), Cholette et al. (2024), and Morrison et al. (2025). In this study, the two-moment ice configuration, with prognostic liquid fraction off, and with 3 ice categories is used.

Hydrometeor size distributions are represented by complete 3-parameter gamma functions, with shape and slope parameters evolving dynamically based on the prognostic variables. The configuration uses the two-moment treatment of ice particles instead of the triple-moment treatment which is more relevant for deep convective systems to better resolve size sorting and improve simulations of hail and heavily rimed particles. Additionally, at the time of preparing this study, the "full" 3-moment version of P3 Morrison et al. (2025) was not yet available. In this version, the spectral dispersion changes due to deposition/-
sublimation which could be a topic of future work for contrails.

To examine the impact of the deposition coefficient ($\alpha_D$, from equation A2) on the mass growth of ice crystals (equation A1) and its subsequent effect on contrail persistence, we conducted sensitivity simulations. These simulations introduce reduction factors ($rf$) to adjust the depositional growth of ice particles heuristically.

$$\frac{\left(\frac{dm}{dt}\right)_{\alpha_D}}{\left(\frac{dm}{dt}\right)_{\alpha_D=1}} = rf, \quad \text{where} \quad rf \in \{0.6, 0.8, 0.9\} \tag{1}$$

The reduction factor $rf$ is determined by varying the ice particle size and computing the corresponding $\alpha_D$. These size-dependent $\alpha_D$ values are then used to calculate the depositional growth rate, which is subsequently compared to the reference case where $\alpha_D = 1$ in the depositional growth rate equation. Further information on this approach can be found in Appendix A.

### 2.2.3   Contrail model: CoAT

The formation of contrails, governed by the Schmidt–Appleman Criterion (SAC), requires specific thermodynamic conditions that depend on both atmospheric and aircraft parameters. A critical parameter in this framework is the slope of the mixing line, $G$, which characterizes the relationship between the exhaust plume and the ambient atmosphere. According to Schumann (2012), $G$ is expressed as:

$$G = \frac{c_p \cdot p \cdot EI_{H_2O}}{\left(\frac{M_{H_2O}}{M_{\text{air}}}\right) \cdot Q \cdot (1 - \eta)}, \tag{2}$$

where $c_p$ is the specific heat capacity of air at constant pressure (1004 J kg$^{-1}$ K$^{-1}$), $p$ is the ambient pressure (Pa), $EI_{H_2O}$ is the water emission index (kg of $H_2O$ per kg of fuel burned), $M_{H_2O}/M_{\text{air}}$ is the molar mass ratio of water to air (approximately 0.622), $Q$ is the specific combustion heat of the fuel (43.2 MJ kg$^{-1}$ for kerosene), and $\eta$ is the propulsion efficiency (0.29).

For contrails to form, the ambient temperature $T$ must be below the maximum threshold temperature, defined as the tem-
perature at which the mixing line intersects the liquid water saturation curve (Schumann, 2012). Additionally, the relative humidity over water must exceed the critical threshold to ensures that the water vapor in the exhaust can condense. Under



these conditions, water vapor condenses and freezes on aerosols, forming ice particles. These ice particles grow by water vapor deposition, and if ice supersaturation persists, the contrails can evolve into long-lived clouds.

The wake vortex phase of contrail evolution involves distinct processes in the primary and secondary wakes, critical for understanding contrail dynamics and their transition into cirrus clouds. The primary wake, associated with the counter-rotating vortex pair generated by the aircraft's lift, experiences strong downward motion, leading to adiabatic heating and partial sublimation of ice particles. These dynamics, which trap a significant portion of the exhaust within the vortex pair, are strongly influenced by environmental factors such as temperature and relative humidity (Sussmann and Gierens, 1999; Unterstrasser, 2016). In contrast, the secondary wake forms a "curtain" of detrained exhaust between the original emission altitude and the descending vortex. Ice particles in the secondary wake grow through deposition in an ice-supersaturated environment and retain the majority of the contrail ice mass by the end of the vortex phase. This secondary wake, less affected by adiabatic heating, plays a crucial role in the persistence and spreading of contrails (Unterstrasser, 2014; Lewellen et al., 2014; Unterstrasser, 2016).

When the conditions for $T$ and $RH_i$ from the SAC are satisfied, GEM uses the wake vortex model from Unterstrasser (2016), which focuses on the interaction between ice microphysics and wake vortex dynamics. The Unterstrasser (2016) parameterization provides a framework for quantifying key characteristics of young contrails during their early development phases. It calculates the maximum vertical displacement of wake vortices and the vertical extent of the contrail, which corresponds to the maximum vortex displacement if ice particles can survive the warming effects associated with the adiabatic descent of the vortices. Additionally, it estimates the survival fraction of contrail ice particles, accounting for their loss due to changes in relative humidity resulting from adiabatic warming and incorporating the influence of ice supersaturation, temperature, and the Kelvin effect on crystal growth (Jensen et al., 2024). The wake vortex model uses the aircraft information from Table 2 and the atmospheric conditions (e.g. temperature, pressure, humidity and the static stability of air) from GEM to determine ice particle survival, the contrail ice number concentration and the vertical contrail extent of young contrails during the first 5 min. The wake vortex model enables parameterizations to incorporate additional aircraft characteristics, such as weight, speed, and fuel flow rate. However, for our purposes, we utilized only the parameterization based on wingspan for further calculations. To estimate the contrail ice mass concentration, the sublimated ice particles from the primary vortex are converted into water vapor and incorporated into the plume's water vapor budget. The surviving ice particles grow through deposition, assuming spherical geometry for the ice particles (Pruppacher and Klett, 2010). The resulting contrail ice number and mass concentrations derived from the wake vortex model are subsequently advected by GEM. If the contrail's vertical extent exceeds that of the model grid, the contrail's ice content is distributed into the model grid box below the flight level (Appendix A2).

## 3 Results

Here, we present two distinct simulation approaches, along with the sensitivity simulations, for analyzing contrail formation. First, we conducted simulations assuming the characteristics of an A321 aircraft and a B747 aircraft uniformly across the entire



domain. This approach allowed us to generate metrics such as contrail depth, contrail ice number concentration (CINC), and ice particle survival (IPS) at each timestep, as if an A321 or B747 were present throughout the domain. This methodology provides insights into potential contrail formation areas within the domain. The outcomes of these simulations are discussed in Sections 3.1.3 and 3.2.1. Second, we performed simulations where the model utilized actual flight data to extract specific aircraft locations and timings. Contrail characteristics were then simulated exclusively along the recorded flight paths, based

on the aircraft properties detailed in Table 2. These findings are elaborated in Section 3.2.2.

### 3.1    Depositional growth sensitivity simulations

### 3.1.1    RHi distribution: Sounding vs GEM

The approach examines the relationship between ice particle growth rates and $RH_i$ by analyzing soundings and comparing them to simulations from the GEM model. The simulations are designed to illustrate how variations in ice particle growth rates

influence $RH_i$. By contrasting the sounding observations with GEM under different growth rate scenarios, we aim to elucidate the impact of ice particle depositional growth on atmospheric humidity profiles. To account for uncertainty in balloon drift, a $30\,\mathrm{km} \times 30\,\mathrm{km}$ area surrounding balloon location while ascending was used to compile the vertical profile for each station in GEM's output, which is then compared to the soundings.

    Figure 2 presents the $RH_i$ distribution for sounding observations compared to GEM at 1200 UTC on 25 Nov 2023, consid-

ering pressure levels between 100 hPa and 600 hPa and temperatures colder than $-38\,^{\circ}\mathrm{C}$. The simulations capture the general distribution well below an $RH_i$ of 100 % when compared to the all-soundings distribution. However, as $RH_i$ approaches 100 %, GEM underestimates the probability of higher $RH_i$ values, particularly in the supersaturation range, where it fails to capture the frequency and intensity of ice supersaturation events. The combined sounding $RH_i$ distribution peaks at 100 % and then gradually decreases, reaching a maximum $RH_i$ of 145 %.

The CNTL simulation follows a trend commonly observed in atmospheric models, where the $RH_i$ distribution peaks around 100 % before sharply declining due to rapid humidity quenching by ice growth schemes (Kärcher et al., 2023) or models using saturation adjustment schemes (Tompkins et al., 2007; Gierens et al., 2020). Unlike models employing a saturation adjustment, GEM's P3 microphysics scheme does not use such a constraint. When the depositional growth rate is reduced to 90 % (Dep_0.9) of the CNTL value, the $RH_i$ distribution peak shifts from 100 % to 104 %, and the maximum $RH_i$ increases from 104 % to

108 %. Further reductions to 80 % (Dep_0.8) and 60 % (Dep_0.6) enhance moisture buildup, leading to higher $RH_i$ maxima of 116 % and 128 %, respectively, which align more closely with observations. However, these reductions also lower the distinct 100 % peak seen in the CNTL, suggesting a redistribution of humidity. The largest contributor to the underestimation is GEM's inability to capture the DTX sounding (Fig. B1).



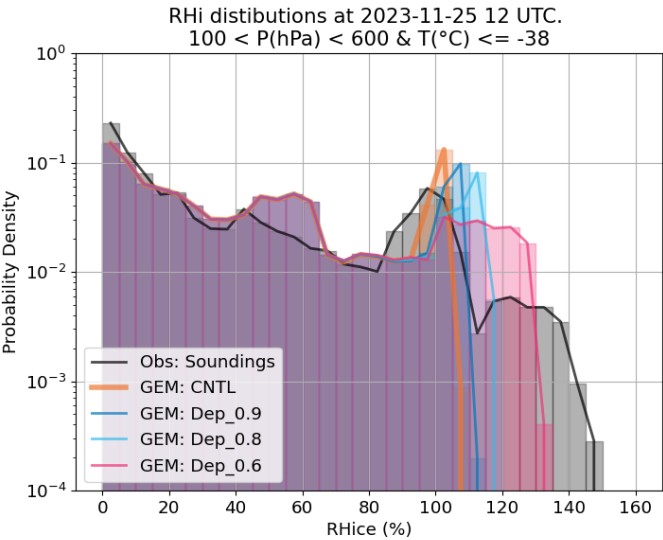

**Figure 2.** RHi distribution for CNTL, Dep_0.9, Dep_0.8 and Dep_0.6 simulations at 1200 UTC. The observations from all the radiosonde sounding is combined and shown as "Obs: Soundings" (black line). The GEM soundings include data in a 5 km × 5 km domain around the location of the balloon to account for uncertainty.



### 3.1.2 The outlier: DTX sounding

At the 500 hPa level, a deep trough was present over the Great Lakes region, indicating an area of lower pressure and cooler temperatures aloft (Fig. 1). This trough was associated with enhanced jet stream activity, which led to increased upper-level divergence. Such divergence promotes rising motion in the atmosphere, generating thin streaks of ice-supersaturated regions, which in turn favored cirrus cloud formation and the potential for persistent contrail development.

The DTX sounding profile reveals that the balloon drifted through one of these supersaturated streaks, recording $RH_i$ val-
ues exceeding 140 % between 300 hPa and 200 hPa. While GEM accurately captured the broader upper-air trough, it failed to generate the highly ice-supersaturated regions above DTX (Fig. 3a). In contrast, most of the other stations, except for ALB, exhibited ice supersaturation between 400 hPa and 200 hPa, which GEM represented better (e.g., Fig. 3b). However, this improved representation may be due to the fact that the other soundings did not contain such thick layers of highly ice-supersaturated air, making them easier for the model to capture.

When the depositional growth rate is slowed the sensitivity simulations show a marginal improvement, with RHi values increasing to 106 % (Dep_0.9), 112 % (Dep_0.8), and 122 % (Dep_0.6). While this adjustment allows for a higher degree of ice supersaturation, the model still underestimates the extreme RHi values observed in the DTX sounding, where RHi exceeded 140 % (Fig. 3a). In comparison, the ice supersaturated regions are much better represented for the BUF sounding, especially in the Dep_0.8 and Dep_0.6 simulations. These slower deposition growth rates corresponds to $\alpha_d$ smaller than 0.1 for ice particle
diameters of larger than 50 μm (Fig. A1). There are very thin ice-supersaturated layers, such as at 220 hPa in Fig. 3b, which are not well represented in the GEM simulations. This discrepancy may be attributed to the vertical resolution of the model, where the layer thickness at this pressure level is approximately 250 m. The model's relatively coarse vertical resolution may smooth out small-scale supersaturation features, limiting its ability to resolve narrow layers of high $RH_i$ observed in the soundings.



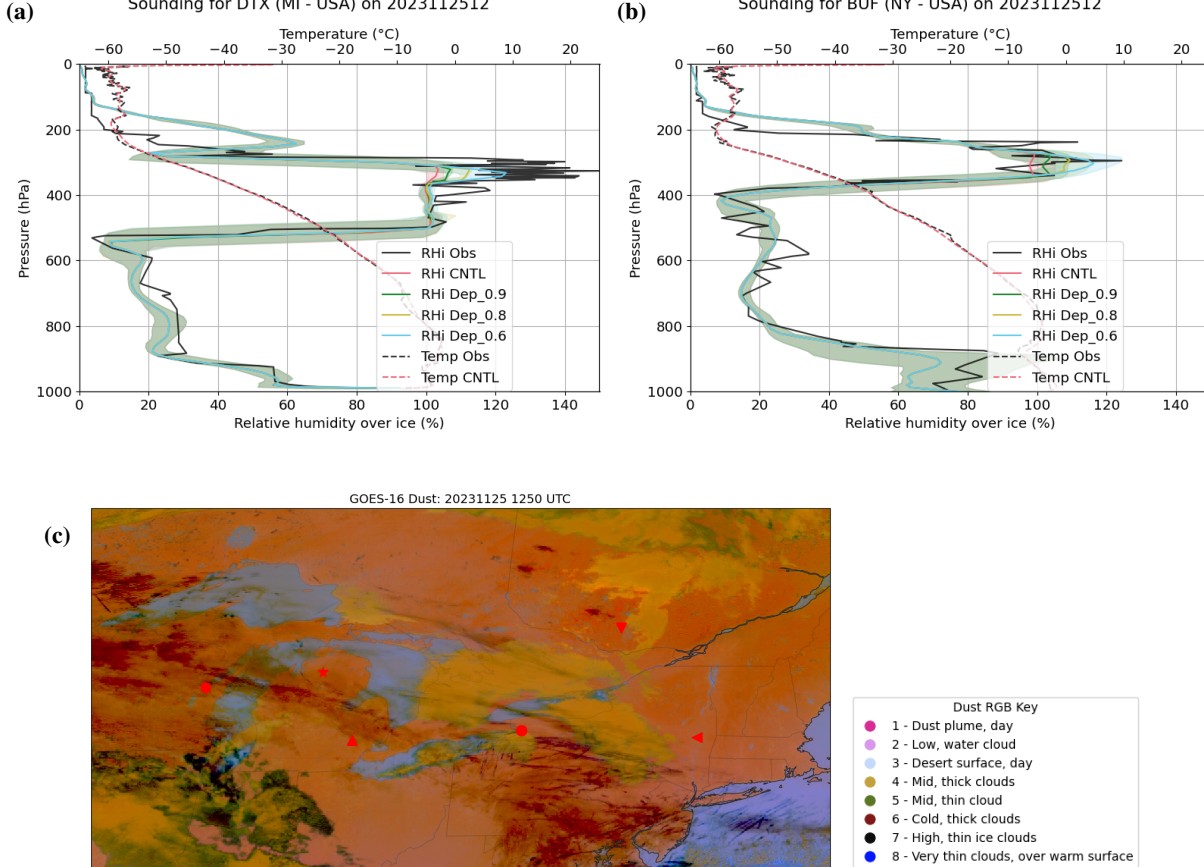

**Figure 3.** Soundings for **(a)** DTX, **(b)** BUF compared to the Dep_0.9, Dep_0.8 and Dep_0.6 simulations. The shaded areas are the minimum and maximum values for a 5 km × 5 km domain around the location of the balloon. Panel **(c)** is the GOES-16 Dust RGB at 1250 UTC





### 3.1.3 GEM vs ceilometer observations at CYYZ

In the Great Lakes region, the interaction between cold Arctic air masses and the relatively warmer lake waters frequently triggers lake-effect snow events (Niziol et al., 1995). The synoptic conditions on 25 Nov 2023 and the following days facilitated such phenomena, leading to the formation of low clouds and heavy snowfall in the surrounding areas. On this day, these low clouds significantly attenuated the ceilometer signal, making it nearly impossible to retrieve data from cirrus clouds between 1300 UTC and 1700 UTC. However, a descending high cloud base was still noticeable, aligning with the descending moist

layer observed in all simulations (Fig. 5).

Between 1100 UTC and 1200 UTC, the ceilometer's backscatter intensity displayed a diffused structure, indicative of thin ice clouds like cirrus or contrails around $\sim 10\,\mathrm{km}$ in altitude. By 1200 UTC to 1300 UTC, the cirrus clouds became more structured, with a lower cloud base near 8 km, suggesting the presence of ice-supersaturated regions. Figure 4 presents a time series of the $RH_i$ profiles for all simulations at the same location as the CYYX ceilometer. The Dep_0.8 and Dep_0.6 simulations support

the presence of ice-supersaturated regions between 8 and 10 km and indicate the potential for persistent contrail formation around $\sim 10\,\mathrm{km}$.

From 1300 UTC until around 1500 UTC, all simulations except Dep_0.6 failed to indicate conditions suitable for persistent contrail formation. During this time, contrails were observed over Toronto, but images at 1420 UTC and simulations at 1400 UTC show no indication of new contrail formation, only widespread older contrails that had formed upstream towards

the west.

Variations in the depositional growth rates of ice particles significantly affect the persistence and spatial extent of ice-supersaturated regions. While all sensitivity simulations indicate the presence of contrail-forming regions, the CNTL simulations exhibit conditions that are excessively dry prior to 1645 UTC. After this time, although ice-supersaturated regions persists above 6 km, the ambient temperature remains too elevated, warmer than $-45\,^{\circ}\mathrm{C}$, for the aircraft plume to achieve supersat-

uration. Consequently, the nucleation and freezing of water droplets, essential for contrail persistence in ice-supersaturated environments, are inhibited. In contrast, the reduced moisture deposition rate, synonymous to lower $\alpha_d$ values, in the sensitivity simulations allows for a more extensive and intense presence of ice-supersaturated regions, thereby enhancing the likelihood of persistent contrail formation.

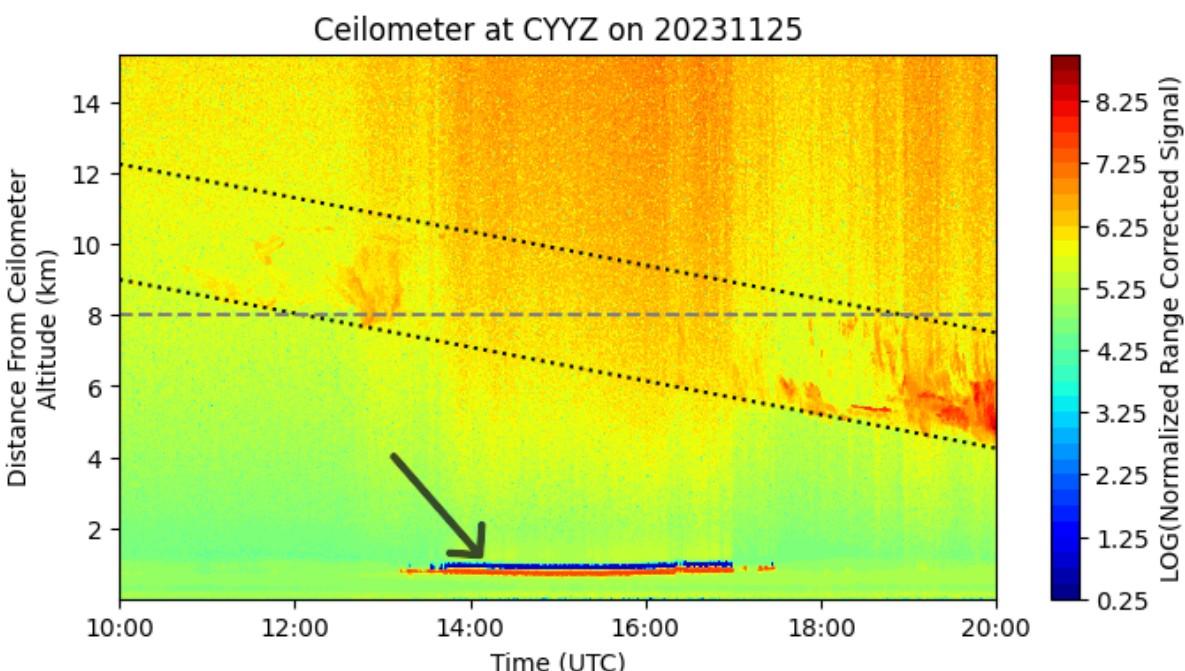

**Figure 4.** Time series of ceilometer backscatter profiles for 25 Nov 2023 at Pearson International Airport (CYYZ, Toronto). The dotted lines indicate the descending cloud bottom and cloud top over time. The dashed line shows the altitude of the cloud bottom at 1200 UTC. The arrow indicates the low-level stratus between 1300 UTC and 1700 UTC.





**Figure 5.** Time series for the RH$_i$ for the **(a)** CNTL, **(b)** Dep_0.9, **(c)** Dep_0.8 and **(d)** Dep_0.6 simulations between 1200 UTC and 1800 UTC. The grey shaded areas are the ice number concentration from ice clouds in the model. The dotted line one the where the temperature is $-45\,^\circ$C and the solid grey lines are the flight levels (FL).



### 3.2  CoAT simulations

#### 3.2.1  Sounding vs CoAT

Here, we compare the atmospheric conditions at three sounding stations (ALB, BUF, and APX) spanning west to east along the main flight corridor at 1200 UTC. The sounding data provide input for the CoAT model, which determines the formation of persistent contrail regions and evaluates contrail properties, such as contrail depth and CINC that survive the wake vortex. These simulations are then compared to results where the simulated atmospheric conditions from GEM serve as input for CoAT. For all comparisons, we assume the aircraft characteristics of an A321 and only consider the Dep_0.8 simulation.

Figure 6 illustrates that flights traveling between ALB and BUF will begin to encounter high ice clouds. At ALB, the upper atmosphere remains dry and is not conducive to cirrus or contrail formation (Fig. 6a). In contrast, at BUF, between 400 hPa and 200 hPa, the observed $RH_i$ exhibits significant variability, with multiple layers exceeding 100 %, indicating the presence of ice-supersaturated regions. This is particularly pronounced between 300 hPa and 260 hPa, as well as in a very shallow layer at 240 hPa, where $RH_i$ frequently surpasses 100 %. In these regions, the ambient temperature is below the critical threshold for SAC, creating favorable conditions for persistent contrail formation (Fig. 6b).

While GEM generally captures the ice-supersaturated regions well, it does not fully resolve the magnitude of the observed supersaturation events, often smoothing out peaks where $RH_i$ exceeds 100 %. This discrepancy suggests that accurately capturing thin ice-supersaturated layers requires higher vertical resolution in models. The presence of highly localized supersaturation spikes (e.g. at 290 hPa) leads to CoAT generating deeper contrail depths and higher CINC of 310 m and 190 cm$^{-3}$, respectively, compared to GEM-CoAT, which produced contrail depths and CINC of 160 m $\in$ [100 m, 275 m] and 60 cm$^{-3}$ $\in$ [58 cm$^{-3}$, 80 cm$^{-3}$], respectively (BUF, Fig. 6b).

It is important to note that in CoAT, SAC regions can be present even when the wake vortex model indicates no contrail formation (i.e., contrail depth and CINC of zero). This occurs because the wake vortex model accounts for adiabatic processes during plume descent, where warming of the air leads to ice particle sublimation. In this case, despite a temperature of $-52\,°C$ and an $RH_i$ of 102.5 % at a pressure of 270 hPa, all ice particles sublimated within the first few minutes of the young contrail's lifetime. This indicates that merely using SAC as an indicator for contrail persistence may overestimate the regions in which ice particles can survive the wake vortex dynamics.

In the case of the APX sounding (see Fig. 6c), the regions of supersaturation with respect to ice (SAC) appear to match reasonably well between the observational data and the GEM-CoAT simulation. However, the figure indicates that GEM-CoAT predicts a higher and more vertically extensive $RH_i$ from approximately 340 hPa to 240 hPa, compared to the narrower layer observed in the sounding data. As a result, GEM-CoAT overestimates both the contrail depth, generally simulating thicknesses between 250 m and 300 m, and the CINC. In contrast, CoAT typically shows contrail depths between 50 m and 100 m, except for one thin layer where the contrail depth reaches 225 m.

In general, an A321 aircraft operating along the ALB-BUF-APX corridor may have generated persistent contrails in the vicinity of the BUF and APX stations between 350 hPa and 250 hPa at 1200 UTC. Subsequently, at approximately 1300 UTC, flight DL384 traversed the Toronto region along the ALB-BUF-APX corridor at an altitude of ~300 hPa (30,000 ft). The




analysis of this specific flight path is superimposed on the output of the model simulations, with the potential for contrail formation discussed in the following section.





**Figure 6.** Atmospheric soundings for **(a)** ALB, **(b)** BUF and **(c)** APX compared to the GEM-CoAT Dep_0.8 simulation. The shaded areas are the minimum and maximum values for a $5\,\mathrm{km} \times 5\,\mathrm{km}$ domain around the location of the balloon. The shaded yellow and hatched areas are where persistent contrails can form according to the Schmidt-Appelman Criterion (SAC) in CoAT (calculated using the sounding information) and GEM-CoAT (calculated using the model data), respectively. The grey lines are the flight levels (FL).





### 3.2.2 Superimposed cross-section flight track analysis

Figure 7 illustrates the simulated cross-sectional regions of persistent contrail formation, highlighting areas where ice particles survive the wake vortex along the A321 flight route. The aircraft, operating along the ALB-BUF-APX corridor, maintains an altitude of 300 hPa (30,000 ft) between $-74\,°$W and $-84\,°$W before ascending to 240 hPa (34,000 ft) at $-84\,°$W, where it continues cruising. Over this cross-section, the CoAT Dep_0.8 simulation predicts young contrails with a CINC mean [10th percentile, 90th percentile] of 83.9 cm$^{-3}$ [62.8 cm$^{-3}$, 118.5 cm$^{-3}$] (Figure 7a and b).

During its cruise at 300 hPa, the A321 traverses regions favorable for contrail formation, particularly over Lake Ontario as it approaches the APX station. Observations from GOES-16 Dust imagery confirm the presence of multiple aircraft-generated contrails between 1300 UTC and 1500 UTC (Figure 1). Near the APX station, the A321 ascends out of the contrail-forming layer into a drier atmospheric region, where contrail formation ceases.

A comparison between the A321 Dep_0.8 (Figure 7b) and A321 CNTL (Figure 7d) reveals that the faster deposition growth simulation (CNTL) exhibits a lower mean CINC (64.6 cm$^{-3}$). This trend is further supported by the significantly lower CINC 10th percentile in the A321 CNTL simulation (5.6 cm$^{-3}$), indicating a shallower distribution of ice particles under faster depositional growth rates. The reduction in ice particle survival is attributed to the enhanced moisture depletion in the ice-supersaturated regions due to faster deposition, leading to a decrease in RH$_i$ and ultimately lowering the ice number survival fraction during the wake vortex phase (Figure 7a and c). Consequently, the A321 CNTL simulation produces shallower contrails (detailed in the appendix) with lower contrail ice concentrations compared to the Dep_0.8 simulation, impacting the contrail's lifetime and evolution.

When the A321 aircraft (medium-weight category) is replaced by a B747 (heavy-weight category) in the B747 Dep_0.8 and B747 CNTL simulations, stark differences emerge. The B747 CNTL simulation indicates a significant reduction in the contrail formation region, suggesting that heavy-weight category aircraft flying the same route and at the same time may form fewer or no contrails compared to the medium-weight category aircraft (Figure 7d and h). The deeper wake generated by the B747 results in increased adiabatic heating and reduced ice particle survival, leading to a mean CINC of only 14.3 cm$^{-3}$ compared to 64.6 cm$^{-3}$ for the A321 (Table 3). A similar, though less pronounced, reduction is observed between the B747 Dep_0.8 (61.1 cm$^{-3}$) and A321 Dep_0.8 (83.9 cm$^{-3}$) simulations.

These findings highlight the role that aircraft weight, especially at near ice-supersaturated conditions, and deposition growth rates play in simulating contrail formation and persistence accurately. While faster deposition reduces contrail ice concentrations, heavier aircraft further amplify this effect by increasing wake turbulence and adiabatic heating, ultimately limiting the conditions necessary for persistent contrails.





**Table 3.** Comparison of the Contrail Ice Number Concentration (CINC) and the Ice Particle Survival (IPS) mean, 10th and 90th percentiles for A321 Dep_0.8, A321 CNTL, B747 Dep_0.8, and B747 CNTL along the flight route in Figure 7.

| Parameter | A321 0.8_Dep | A321 CNTL | B747 0.8_Dep | B747 CNTL |
|---|---|---|---|---|
| CINC mean ($cm^{-3}$) | 83.9 | 64.6 | 61.1 | 14.3 |
| CINC 10th ($cm^{-3}$) | 62.8 | 5.6 | 0.0 | 0.0 |
| CINC 90th ($cm^{-3}$) | 118.4 | 87.8 | 88.3 | 65.8 |
| IPS mean (%) | 12.1 | 2.4 | 7.0 | 1.0 |
| IPS 10th perc (%) | 2.3 | 0.5 | 1.2 | 0.1 |
| IPS 90th perc (%) | 22.1 | 4.7 | 12.4 | 2.2 |



**Figure 7.** Vertical cross-section along the flight path (ALB-BUF-APX) for **(a)** DL384 Dep_0.8 (A321), **(b)** DL384 CNTL (A321), **(c)** DL384 Dep_0.8 (B747) and **(d)** DL384 Dep_0.8 (B747) overlaid over regions showing the ice particle survival and contrail ice number concentration at 1300 UTC.





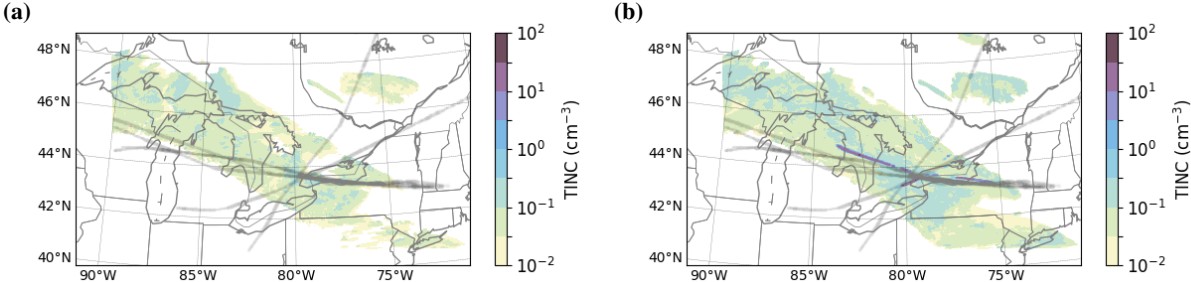

**Figure 8.** Combined ice number concentration from GEM-P3 nucleation scheme and the GEM-CoAT scheme. The maximum ice number concentration is selected between 320 hPa and 200 hPa and $RH_i$ larger than 90 % and plotted. Flight information from Table 2 is used as input for the **(a)** CNTL and **(b)** Dep_0.8 simulations at 1420 UTC. The flight tracks are shown in grey lines.

### 3.2.3 Contrail simulations in GEM-CoAT

In this section, we incorporate the flight information into GEM-CoAT using the aircraft characteristics listed in Table 2. We simulate contrail formation based on the variation of aircraft properties and flight routes.

From the GOES-16 observations and photographs (Fig. 1), it is evident that multiple aircraft-generated contrails were present over the Toronto area at 1420 UTC. Contrail formation began around 1200 UTC, increasing in extent and number until 1600 UTC. Here, we analyze the simulated contrails formed by three specific aircraft: TK6061 (B747), DL384 (A321),

and CV6686 (B747). The CINC produced by these aircraft is combined with the ice number concentration (INC) generated by GEM-P3's nucleation schemes, to get the total ice number concentration (TINC), before being advected. The simulated contrails are then compared with GOES-16 Dust observations at 1420 UTC. Figure 8a highlights the limitation of using standard depositional growth rates, as the CNTL simulation fails to reproduce any contrails over the domain at any time. Instead, it only represents INC from GEM-P3's ice nucleation scheme. In contrast, the Dep_0.8 simulation successfully captures contrail

formation, particularly from the TK6061 and DL384 aircraft, which were flying at flight levels FL310 and FL300, respectively (Fig. 8b and Table 2). These aircraft overflew Toronto at 1345 UTC and 1245 UTC, respectively, while BUF soundings at 1200 UTC indicated persistent contrail formation regions between FL290 and FL330 (Fig. 3b). Additionally, CV6686 briefly formed a contrail north of Toronto, which dissipated within an hour due to its altitude at FL340, where it flew in and out of an ice-supersaturated layer. The locations of these simulated contrails from these aircraft align well with the contrails detected by

GOES-16 Dust observations, a result that is only achieved when depositional growth rates are slowed.




## 4 Summary and conclusions

On 25 November 2023, between 1200 UTC and 1800 UTC, widespread ice-supersaturated regions formed, resulting in a high occurrence of contrail formation over eastern Canada and the USA. Photographic images from Toronto and satellite-based observations from the GOES-16 Advanced Baseline Imager Dust Red-Green-Blue composite indicated that aviation contrails persisted for several hours, especially over the Lake Ontario region. Ceilometer data taken from Toronto Pearson International Airport (CYYZ), radiosonde soundings from Albany, Gaylord, Buffalo, White Lake, Green Bay, International Falls, Maniwaki, and Pickle Lake were used to analyze the atmospheric conditions under which the ice-supersaturated regions formed.

The Global Environmental Multiscale (GEM) model, which includes the Predicted Particle Properties (P3) microphysics scheme was employed as the base model for high-resolution simulations. The Contrail Avoidance Tool (CoAT), which consists of Schmidt-Appleman Criteria (SAC) and the wake vortex model from, was used to determine persistent contrail forming regions and contrail properties.

First, we analyzed the ability of GEM-P3 to simulate ice-supersaturated regions by heuristically adjusting the depositional growth rate of ice particles and to compare it against the CNTL simulation, where no reduction factor was applied. The findings can be summarized as follows:

- The CNTL simulation underestimates $RH_i$ distribution, following a common trend in atmospheric models in which the moisture is quenched too quickly, resulting in a $RH_i$ peak of $\sim100\,\%$.

- Sensitivity simulations indicate that reductions in ice particle depositional growth rates enhances moisture buildup, leading to improved forecasts of $RH_i$ influencing the extent of contrail forming regions.

- In CoAT, the presence of persistent contrail forming regions (SAC) does not necessarily imply contrail formation, as wake vortex dynamics can induce adiabatic warming during plume descent, leading to complete ice particle sublimation despite favorable thermodynamic conditions. This suggests that SAC alone may overestimate the spatial extent of persistent contrail regions.

Second, the CoAT model was employed to simulate regions of persistent contrail formation and associated microphysical properties for two aircraft types: the A321 (medium-weight) and the B747 (heavy-weight), along the ALB-BUF-APX flight corridor. The key findings are summarized as follows:

- In the Dep_0.8 simulations, the B747 induces deeper wake vortices, leading to enhanced adiabatic heating and a subsequent 27 % reduction in mean contrail ice number concentrations relative to the A321.

- In the CNTL simulations, this reduction is even more pronounced at 78%, suggesting that heavier aircraft may significantly inhibit contrail formation under similar atmospheric conditions due to increased wake turbulence and ice particle sublimation.

These results underscore the critical influence of aircraft-specific characteristics on contrail formation and persistence. More importantly, they demonstrate that with a reduced depositional growth rate of ice particles, the GEM-CoAT model was able





to reproduce the observed contrails.This highlights the need to accurately represent ice supersaturation processes in numerical simulations to improve the fidelity of contrail modeling.

In this context, recent work on the introduction of the 3-moment treatment of ice in P3 by Milbrandt et al. (2021) has advanced the representation of microphysical processes, with P3 now fully 3-moment for all ice-related processes (Morrison et al., 2025). This allows the shape parameter of the ice size distribution, which is proportional to the relative spectral dispersion, to evolve independently for all processes, including depositional growth. As a result, the deposition rate both influences and is

420 influenced by the shape parameter. Incorporating these interactions should, in principle, lead to improved representation of ice-supersaturated regions in the upper troposphere within GEM-P3. The underestimation of ice supersaturation in models can be attributed to limitations in their representation of phase relaxation, turbulence, and layer resolution. Korolev and Mazin (2003) shows the important role of phase relaxation in regulating supersaturation within ice clouds. In typical cirrus conditions, where the ice crystal number concentration and size are approximately $20\,\mathrm{cm}^{-3}$ and $0.2\,\mu\mathrm{m}$, respectively (Lohmann et al., 2016), the

425 phase relaxation timescale is around $200\,\mathrm{s}$. These timescales can range between $60\,\mathrm{s}$ and $460\,\mathrm{s}$ in cirrus clouds, exceeding the timesteps commonly used in models. The substantially longer phase relaxation timescales allow supersaturation to persist beyond what is typically represented in numerical models. Our CNTL simulation in GEM-P3 is unable to reproduce the buildup of supersaturation while using a $30\,\mathrm{s}$ model timestep, unless the ice deposition growth rate is adjusted. A limitation in the P3 microphysics scheme used in this study is its treatment of ice nucleation. While the scheme has undergone several improve-

430 ments, the deposition nucleation process forms ice even at temperatures colder than $-38\,^{\circ}\mathrm{C}$, for which it was not originally designed. Additionally, the ice nucleation rate is currently set to $0.1\,\mathrm{cm}^{-3}\mathrm{s}^{-1}$, resulting in ice relaxation timescales of approximately $340\,\mathrm{s}$, toward the lower end of the range associated with cirrus clouds. This may contribute to the model's inability to sustain elevated supersaturation levels. Ongoing work aims to refine the representation of ice nucleation at temperatures below $-38\,^{\circ}\mathrm{C}$, following approaches similar to those of Gasparini et al. (2025).

Another factor influencing the underestimation of ice supersaturation is the vertical grid spacing. Many global and regional climate models employ relatively coarse vertical layers, which limit their ability to resolve fine-scale turbulence and small-scale vertical motions that sustain localized supersaturation events. In coarse-resolution models, turbulence-induced variations in humidity tend to be smoothed out, leading to an under representation of extreme supersaturation values (Burkhardt and Kärcher, 2009). This limitation becomes more pronounced when the Richardson number exceeds 0.25, which represents the

ratio of buoyant energy to shear kinetic energy and determines the dynamic stability of the atmosphere (Stull, 2016). In such cases, significant wind shear within well-stratified layers can reduce the persistence of supersaturated regions (Thompson et al., 2024). Although GEM-P3 employs a vertical grid spacing of $230\,\mathrm{m}$ in the upper atmosphere, our CNTL simulations cannot reproduce the very shallow layers of elevated $\mathrm{RH_i}$. This suggests that even relatively high-resolution models may struggle to capture the fine-scale structure of supersaturation layers, potentially contributing to the underestimation bias.





*Code and data availability.* Code will be made available after acceptance but can be showed to reviewers if desired. All data including those used to initialize the simulations, the simulated outputs, used in this study are archived internally for 5-year at the Canadian Meteorological Centre. The plotting software, GEM settings files and CoAT source code can be found at https://doi.org/10.5281/zenodo.15611592. The ceilometer and sounding data (University of Wyoming web site, http://weather.uwyo.edu/) can be found at https://doi.org/10.5281/zenodo.15643030.



# Appendix A

## A1 Depositional growth of ice particles

$$\frac{dm}{dt} = \frac{4\pi C (S_{v,i} - 1)}{\frac{\rho_i RT}{e_{\text{sat},i} D'_v M_w} + \frac{L_s \rho_i}{k'_a T}\left(\frac{L_s M_w}{RT} - 1\right)}, \tag{A1}$$

The equation describes the rate of mass growth of an ice particle, governed by the balance of vapor diffusion and heat conduction. The terms include the geometric capacitance $C$, supersaturation over ice $S_{v,i}$, ice density $\rho_i$, and saturation vapor pressure over ice $e_{\text{sat},i}$, along with the universal gas constant $R$, absolute temperature $T$, and water's molar mass $M_w$. The latent heat of sublimation $L_s$, thermal conductivity $k'_a$, and modified vapor diffusivity $D'_v$ account for heat and mass transfer limitations in the crystal's growth process (Pruppacher and Klett, 2010).

$$D'_v = \frac{D_v}{\frac{r}{r+\Delta_v} + \frac{D_v}{r\alpha_D}\sqrt{\frac{2\pi M_w}{RT_s}}}, \tag{A2}$$

This correction to the standard diffusivity $D_v$ incorporates only the kinetic effects and excludes ventilation, which is negligible for small ice crystals and thus disregarded (Pruppacher and Klett, 2010; Gierens et al., 2003). The key parameters in the kinetic correction factor include the crystal radius $r$ and the "jump" distance $\Delta_v$, typically set equal to the molecular mean free path. $T_s$ represents the surface temperature of the growing ice crystal, and $\alpha_D$ is the deposition coefficient. $\alpha_D$ is defined via the transcendental equation:

$$\alpha_d = \left(\frac{s_{\text{sfc},d}}{s_{\text{crit}}}\right)^b \tanh\left[\left(\frac{s_{\text{crit}}}{s_{\text{sfc},d}}\right)^b\right] \tag{A3}$$

which determines how efficiently water vapor deposits onto ice crystals. It depends on the local ice supersaturation at the crystal surface $s_{\text{sfc},d}$, the critical supersaturation $s_{\text{crit}}$, and a growth mechanism parameter $b$, which controls the transition between different crystal growth modes. The ice supersaturation $s_{\text{sfc},d}$ (Lamb and Verlinde, 2011) at the ice crystal surface and the supercooling function $s_{\text{crit}}$ based on the analysis of Zhang and Harrington (2014) and used by Kärcher et al. (2023)

$$s_{\text{sfc},d} = s\left(1 + \frac{\alpha_d r}{\ell}\right)^{-1}, \quad s_{\text{crit}}[\%] = 0.019655 \cdot (\Delta T[K])^{1.4305} \tag{A4}$$

where $s$ is the ambient supersaturation and $\ell$ the diffusion length. Similar to Kärcher et al. (2023) we define the transition growth regime with a size-dependent growth parameter $m$ for spherical ice crystals:

$$b = \begin{cases} 1, & r < 10\mu m \\ 1 + 14\left(\frac{r - 10\mu m}{70\mu m - 10\mu m}\right), & 10\mu m \leq r \leq 70\mu m \\ 15, & r > 70\mu m \end{cases} \tag{A5}$$





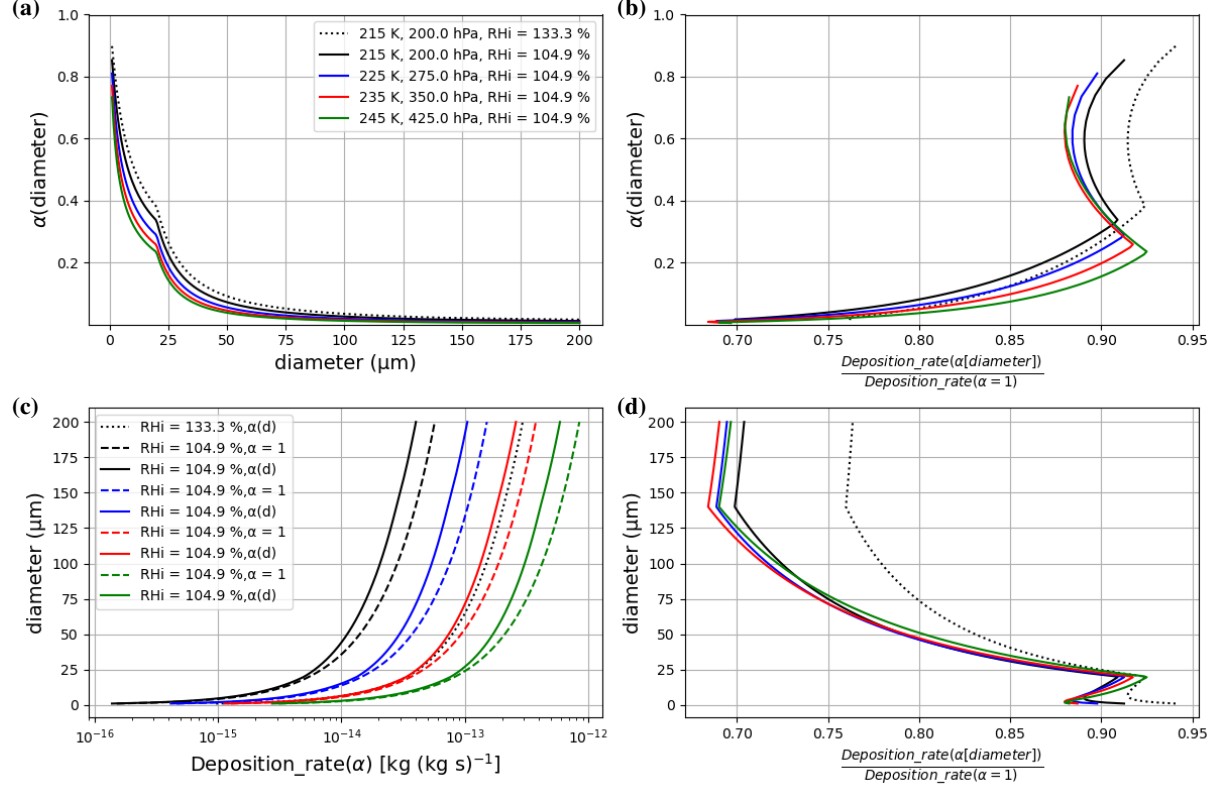

**Figure A1. (a)** The deposition growth coefficient $(\alpha_D)$ as a function of ice particle diameter, **(b)** relationship between $\alpha_D$ and the normalized deposition rate at $\alpha_D = 1$, **(c)** deposition rate as a function of $\alpha_D$ and **(d)** the ice particle diameter dependence on the normalized deposition rate under varying atmospheric conditions.

In P3, the growth of ice particles follows the same formulation as in equation A1, but instead of using $D'_v$, the uncorrected diffusivity is used $D_v$, which is a function of temperature $T$ and pressure $P$ (Hall and Pruppacher, 1976). Consequently, we do not have an explicit description for $\alpha_D$ that can be directly modified to determine the depositional growth rate.

To address this, we solved for $\alpha_D$ for ice crystal sizes between 1 and 200 µm using the Newton-Raphson method, an iterative numerical technique for finding the root of a nonlinear equation and making an initial guess for $\alpha_D = 0.1$ (equations A3 and A4). The computed $\alpha_D$ values were then compared to a reference case where $\alpha_D = 1$, allowing us to evaluate the ratio between $\left(\frac{dm}{dt}\right)_{\alpha_D}$ and $\left(\frac{dm}{dt}\right)_{\alpha_D=1}$.

As shown in Figure A1, this ratio, referred to as the reduction factor for the depositional growth rate, ranges between 0.65 and 0.95 (Fig. A1). Therefore, we implemented reduction factors of 0.6 (Dep_0.6), 0.8 (Dep_0.8) and 0.9 (Dep_0.9) in our sensitivity simulations and compared them to the reference case (CNTL) where $\left(\frac{dm}{dt}\right)_{\alpha_D} = \left(\frac{dm}{dt}\right)_{\alpha_D=1}$.





## A2 Contrail spread over multiple vertical levels

The fractional distributions, $f_{r1}$ and $f_{r2}$, represent how the contrail ice is spread over multiple model levels. Specifically, $f_{r1}$
is the fraction of contrail ice within the current model level, while $f_{r2}$ accounts for the fraction extending beyond the current level into a lower model layer. These fractions are computed using the depth of the contrail ($\mathrm{Cd}(k)$) and the depth of the model level ($\mathrm{Zd}(k)$).

The total ice crystal number and ice crystal mass concentrations in a grid box are denoted by $N_{i_{\mathrm{tot}}}(k)$ and $Q_{i_{\mathrm{tot}}}(k)$, respectively. These quantities are updated by adding the contributions from contrail ice ($N_{i_{\mathrm{contrail}}}(k)$ and $Q_{i_{\mathrm{contrail}}}(k)$) scaled by
the fractional distribution $f_{r1}$. Similarly, the ice content and ice crystals in the level below the current one, represented by $N_{i_{\mathrm{tot}}}(k-1)$ and $Q_{i_{\mathrm{tot}}}(k-1)$, are updated based on the portion of contrail ice extending downward, governed by $f_{r2}$.

$$f_{r1} = \frac{\mathrm{Zd}(k)}{\mathrm{Cd}(k)}, \qquad\qquad f_{r2} = \frac{\mathrm{Cd}(k) - \mathrm{Zd}(k)}{\mathrm{Cd}(k)} \tag{A6}$$

$$N_{i_{\mathrm{tot}}}(k) = N_{i_{\mathrm{tot}}}(k) + N_{i_{\mathrm{contrail}}}(k) \cdot f_{r1}, \qquad\qquad Q_{i_{\mathrm{tot}}}(k) = Q_{i_{\mathrm{tot}}}(k) + Q_{i_{\mathrm{contrail}}}(k) \cdot f_{r1} \tag{A7}$$

$$N_{i_{\mathrm{tot}}}(k-1) = N_{i_{\mathrm{tot}}}(k-1) + N_{i_{\mathrm{contrail}}}(k) \cdot f_{r2}, \qquad\qquad Q_{i_{\mathrm{tot}}}(k-1) = Q_{i_{\mathrm{tot}}}(k-1) + Q_{i_{\mathrm{contrail}}}(k) \cdot f_{r2} \tag{A8}$$

**Appendix B**

## B1 Relative humidity distribution for radiosonde stations

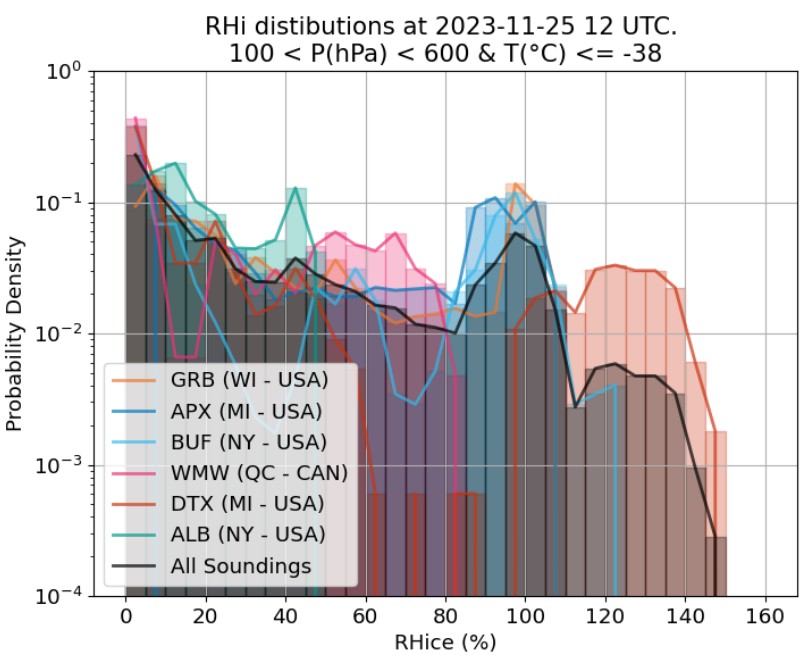

**Figure B1.** Relative humidity with respect to ice distribution for all the radiosonde stations. The distributions only include data between pressures of 600 hPa and 100 hPa, and temperatures colder than $-38\,°C$.



*Author contributions.* ZD conducted the simulations and analyzed the results. ZD was the main author of the paper. AK, JM, ZD contributed to the study's design and the analysis of the results. All authors contributed to the study's writing.

*Competing interests.* The authors declare that they have no conflict of interest.

*Acknowledgements.* ZD acknowledge funding from Transport Canada under agreement STF22-010, and support from Environment and Climate Change Canada's Environmental Protection Branch. ZD also acknowledges and thanks FlightRadar24 for the usage of their proprietary data.





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
