# Peer review of "Improving Forecasts of Persistent Contrails through Ice Deposition Adjustments"

_EGUsphere, 2025_

## Referee Comment (RC2)

**Synopsis**

This manuscript presents a study that is motivated by better forecasting contrail occurrence and deals with improving the prediction of ice-supersaturated regions (ISSR) in an NWP framework. It is suggested to adjust the deposition coefficient, which modifies the life cycle of natural cirrus and hence ice supersaturation in the GEM-P3 model. A variation of this parameter to better forecast ISSRs is novel and interesting.

The study also improves several aspects of existing studies dealing with the same topic. For example, contrail formation is not implemented as a simple binary decision based on the SAC but includes an estimate of the contrail ice crystal number considering ice crystal loss during the vortex phase.

I can recommend publication after the following comments are suitably addressed.

**Major comments**

1. From my perspective, the manuscript has two separate topics:
    1. Better forecasting relative humidity and ice supersaturation by adjusting the deposition coefficient. As a consequence, contrail-forming areas are better predicted, which is a valuable goal.
    2. A case study comparing contrail occurrence in model and multi-faceted observations.

Both topics are relevant, but it should be made clearer that both topics are somewhat independent of each other. For the first topic, a less detailed description of contrails would suffice to identify ice-supersaturated regions. What's your opinion about my perspective?

2. It is not clear how you define contrail persistence. It seems that contrails that have enough ice crystals at the end of vortex phase are called persistent? In GCMs or in contrail plume models, contrails are typically initialized after the vortex phase. Hence, 5-min old contrails may be considered being part of an extended formation stage. Then, persistence relates to contrails with larger lifetimes (let's say e.g. at least 60min, but choosing a precise threshold is certainly subjective). Similarly, around line 80 you state that contrail formation covers the vortex phase. This would contradict your persistence definition ($t_{age}$ being larger than around 5min). Please define formation and persistence and use it consistently. Check also line 399, it may be beneficial to describe in the summary section again precisely what is meant with formation and persistence. Typically, SAC only refers to temperature-controlled formation criterion and not really to persistence in ice-supersaturated regions. Please, be verbose that every reader knows exactly what is meant.

3. A)
    You determined an optimal deposition coefficient for one synoptic scenario: How universally valid is this value? Using GEM-P3 for other synoptic scenarios, would you obtain a similar value? Is your optimal value also relevant for other microphysical models, or do you consider it to be only a tuning parameter of your P3 model?
    B)
    Please also state explicitly that your contrail initialization does not depend on the deposition coefficient, i.e. your CoAT model is independent of $\alpha_d$. Both, ice crystal formation and ice crystal loss during the vortex phase may, however, depend on $\alpha_d$

but this is neglected in the parametrisation of Unterstrasser (2016). Lewellen (2014) looked at a variation of the deposition coefficient.

4. The deposition coefficient is varied only implicitly, see description Appendix A1. I wonder whether it wasn't technically possible to directly vary the parameter within P3. This would simplify the presentation of the results and remove the uncertainty by the non-ideal relation between deposition coefficient and mass growth rate for differently sized ice crystals (e.g. the appendix would be superfluous).

5. I am not sure whether all relevant information is given to understand how contrails are initialized in your model. You mention Unterstrasser (2016), which provides a parametrization of ice crystal survival fraction. What's your choice of the initial ice crystal number (prior to loss)? Does this number depend on ambient conditions? (for ambient temperature close to the SAC temperature, not all soot particles are activated (Lewellen, 2020, Bier et al, 2022)). How do you determine number concentrations n, which requires the specification of a contrail cross-section? How do you determine the ice mass concentration m? Do you specify a mean mass to obtain m from n? How are contrails initialized in subsaturated air, for which Unterstrasser (2016) makes no prediction? Moreover, Unterstrasser (2016) assumes an ice-supersaturated layer with a constant RHi value. As a consequence, the parametrization was not made for shallow layers where the wake vortices move out of the moist region and for which it seems you apply it.
At the end of section 2.2.3 you write that the surviving ice crystals grow through deposition. Due to the very short paragraph, I am not sure whether you really simulate contrails beyond the vortex phase? Then I would assume that you had to feed the contrail properties into P3, which is however not explicitly stated. You also write that GEM advects the number and mass concentrations? Are the contrail ice crystals treated as passive tracers? Or do you really include microphysical processes? Could you please clarify this.

**Minor comments**

1. The title of the study could be refined to „Improved forecast of persistent contrail occurrence …", because you do not simulate persistent contrails
2. line 52: turbulence is typically not a crucial process. Shear and sedimentation are much more important for contrail spreading, see Lewellen, 2014 or Unterstrasser & Gierens, 2010.
3. I do not understand why precise time periods and drift distances are provided for 300hPa.
4. The definition of G is introduced in Schumann 1996, not in Schumann 2012.
5. Line 211: you may cite Lewellen 2012 (instead of Jensen 2024)
6. Fig 2 says 5km, but the text states 30km. What is true?
7. Line 317: what are the numbers in the square brackets?
8. Is correct that $\alpha_d$ is implicitly defined by Eqs. A3 and A4?
9. Section A1 uses radius, Figure A1 uses diameter. Moreover, the axis titles are not "nice" (long, different styles etc.).
10. In general, I believe the font sizes in the figures are too small.
11. Line 55: Unterstrasser (2016) provides a parametrization of early contrail properties. Contrail-cirrus simulations are presented in Unterstrasser et al, 2017a,b or Lewellen, 2014a,b
12. Line 421 and 427: It depends which models you talk about. In typical LES, no phase relaxation (saturation adjustment) is used.

13. The parametrization of Unterstrasser (2016) has been applied in regional and global-scale models. You may find similar findings on ice crystal loss in those studies (Gruber et al, 2017 and Bier & Burkhardt, 2022)

**Technical corrections**

14. A temperature cannot be warm, cooler or colder. This is a feature of the air mass. Temperature is low/high. Similarly, a rate is not fast, only the process that is described.
15. I would not refer to wake vortices as wake turbulence. Once the vortices break up and no coherent dynamical structures exist, then the elevated turbulence intensity might be called wake turbulence.
16. I believe you forgot to delete the sentence in lines 191 & 192.
17. Fig.7 should zoom into the 200 to 400hPa layer. In the present form, most areas in the panels are just white and wasted space.
18. Make sure to use the same subscript: you use $\alpha_d$ and $\alpha_D$.
19. I prefer a mathematical correct notation in eq. A4. One can use units in equations ($s\_crit/\% = \ldots \Delta T/K$).
20. Fig. A1 should extend the y-axis down to 0.0 in all panels.
21. Please check your reference section thoroughly; e.g. $CO_2$ in line 562
22. Line 150: Why is a refinement of the tropical troposphere relevant in your study?
23. Line 390: from STUDY XX?
24. Line 405, for the summary section, it might be useful to explain again what Dep_0.8 refers to.
25. Line 424: Should this be 0.2 mm = 200μm?

**References**

Only those papers are listed that do not appear in your manuscript.

Bier, A., & Burkhardt, U. (2022). Impact of parametrizing microphysical processes in the jet and vortex phase on contrail cirrus properties and radiative forcing. *J. Geophys. Res.*

Gruber, S., Unterstrasser, S., Bechtold, J., Vogel, H., Jung, M., Pak, H., & Vogel, B. (2018). Contrails and their impact on shortwave radiation and photovoltaic power production – a regional model study. *Atmos. Chem. Phys.*, *18*(9), 6393–6411. https://doi.org/10.5194/acp-18-6393-2018

Lewellen, D. C. (2012). Analytic solutions for evolving size distributions of spherical crystals or droplets undergoing diffusional growth in different regimes. *J. Atmos. Sci.*, *69*, 417–434.

Lewellen, D. C. (2014). Persistent contrails and contrail cirrus. Part 2: Full Lifetime Behavior. *J. Atmos. Sci.*, 4420–4438. https://doi.org/10.1175/JAS-D-13-0317.1

Lewellen, David C. (2020). A Large-Eddy Simulation Study of Contrail Ice Number Formation. *J. Atmos. Sci.*, *77*(7), 2585–2604. https://doi.org/10.1175/JAS-D-19-0322.1

Unterstrasser, S., & Gierens, K. (2010a). Numerical simulations of contrail-to-cirrus transition - Part 1: An extensive parametric study. *Atmos. Chem. Phys.*, *10*(4), 2017–2036. https://doi.org/10.5194/acp-10-2017-2010

Unterstrasser, Simon, Gierens, K., Sölch, I., & Lainer, M. (2017). Numerical simulations of homogeneously nucleated natural cirrus and contrail-cirrus. Part 1: How different are they? *Meteorol. Z.*, *26*(6), 621–642. https://doi.org/10.1127/metz/2016/0777

Unterstrasser, Simon, Gierens, K., Sölch, I., & Wirth, M. (2017). Numerical simulations of homogeneously nucleated natural cirrus and contrail-cirrus. Part 2: Interaction on local scale. *Meteorol. Z.*, *26*(6), 643–661. https://doi.org/10.1127/metz/2016/0780

---

## Author Comment (AC1)

**Improving Forecasts of Persistent Contrails through Ice Deposition Adjustments**

Zane Dedekind[1], Alexei Korolev[1], and Jason A. Milbrandt[1]

[1]Meteorological Research Division, Environment and Climate Change Canada, Toronto, Ontario, Canada

**Correspondence:** Zane Dedekind (zane.dedekind@ec.gc.ca)

We sincerely thank Reviewer 1 for the constructive feedback. The suggestions and comments improved the quality of the manuscript.

We have substantially revised the preprint in response to the reviewers' comments, incorporating two main updates:

1. A full solution for calculating the deposition coefficient-based ($\alpha_D$-based) reduction factor has been implemented. $\alpha_D$ depends on temperature, pressure, humidity, and ice particle radius. These are now referred to as the deposition-adjusted (DA) simulations, replacing the previous sensitivity experiments Dep_0.6, Dep_0.8, and Dep_0.9, which are no longer required (Suggested by reviewer 2).

2. The analysis has been extended beyond a single emission index (EI = $1 \times 10^{15}$) to a range of EI values representing soot-rich to soot-poor regimes, allowing assessment of their impact on contrail evolution (Suggested by reviewer 1).

Below we present a detailed response with the reviewer comments in black, our responses in blue and additions to the manuscript in blue italics.

1. The topic of the deposition or accommodation coefficient has been discussed often in the literature. The accommodation factor is certainly a relevant parameter. However, there are also other effects which could be important: This includes, e.g., the number of ice particles (ice nucleation) and assumptions on sub-grid scale variability.
   This is correct; however, our methodology is applied within the context of numerical weather prediction models, where the deposition coefficient is typically set to unity. To address the important issue of accurately simulating upper-tropospheric humidity and, consequently, persistent contrails, we have included an explicit calculation of the deposition coefficient in this study.

2. In respect to the SAC criterium I have a specific remark: The paper concludes among others that the "CoAT simulations revealed that SAC alone is insufficient". It is not clear for what part of the SAC criterium this applies. Please note: It is well known that the SAC criterium does not guarantee the persistence of the contrail. It only decides on contrail formation. So, any warming of the ambient air, e.g. by sinking in the wake vortex, affects the survival of the contrail. This is not an issue of the SAC criterium. This part needs to reformulated.

That is correct and we take the criticism and we removed the statement which was not well phrased. What we intended to convey is that our simulations show cases where, even when the SAC criterion and ice supersaturation were satisfied, contrails did not form—particularly under slight ice-supersaturated conditions and for heavy aircraft such as the B747. We attribute this to differences in wake vortex characteristics between aircraft types. When the relative humidity over ice exceeded 102 %, persistent contrails and contrail-forming regions developed for both the A321 (medium-weight category) and the B747 (heavy-weight category).

3. Is the GEM (as the name suggests) a global model? Or is it a limited area model (as indicated by the information on page 7, lines 148 ff)?

   This can be confusing. GEM is a global modeling framework developed by Environment and Climate Change Canada that can be configured for different spatial scales. In the HRDPS system, GEM is used in a high-resolution configuration (2.5 km grid spacing) for regional weather prediction over Canada. We use these generated 2.5 km output fields, as referenced in our manuscript, to drive our regional 1 km × 1 km resolution simulations.

4. The description of P3 uses the term "property-based approach" (line 157). I do not know what an property based approach is. So, it seems, I have to read all the references given? Line 164 says "with prognostic liquid fraction off" – does this mean the model works without treating liquid water? Why is this a critical assumption for this application and why did you need to menton it?

   In the P3 (Predicted Particle Properties) microphysics scheme, the particle property approach means that instead of using fixed categories for different hydrometeors (e.g., cloud ice, snow, graupel), the model predicts key physical properties of ice particles — such as mass, number concentration, bulk density, and rime fraction — directly through prognostic equations. This allows the model to evolve particle characteristics continuously based on environmental conditions, rather than switching between discrete species. In other words, particle behavior emerges from their predicted properties, not from pre-defined categories. We made some slight modifications to the description:

   *The P3 microphysics scheme is unique in that the ice phase uses the property-based approach, in contrast with the traditional approach of predefined ice-phase categories (e.g. ice, snow, graupel and hail), whereby all ice-phase or mixed-phase particles are represented by one or more generic or "free" categories whose bulk physical properties—such as mass, number, density, and rime fraction—evolve freely and continuously.*

5. The P3 model within GEM is applied using 3 ice categories (line 164). Which are these categories? How are the outputs of CoAT (lines 218/219) related to these ice categories?

   In the GEM-P3 setup with three ice categories, the model predicts three distinct but physically consistent ice populations whose bulk properties evolve freely, allowing representation of various ice types (e.g. cloud ice, snow, graupel) without prescribing fixed categories. In this study, contrail ice crystals from CoAT are initialized in the first P3 ice category, representing small, pristine ice particles.

6. Why is figure B1 in an appendix, which contains nothing else than just this figure? In this figure, the various radiosonde contributions are hard to distinguish. I see red and blue colors but the rest is just in a color mix which I cannot discriminate. Moreover, the figure is hard to read because I am unfamiliar with the various Radiosonde names and their positions (GRB etc.). Which radiosonde shows the results for the airport of Toronto? Where in your map (Fig 1 a) is Toronto? By the way, Fig 1a is not referenced in the text. Line 253 says the "largest contribution to the underestimation is GEM's is its inability to capture the DTC sounding (Fig- B1)". I cannot understand this by only looking to the figure B1. Please provide further explanations (without abbreviations).

We have simplified the figure to improve readability and added the locations of all sounding stations in Figs. 1a, 1b, and 3c. Where relevant, station names are now spelled out in the captions. Table 1 provides the full list of stations, including their names, locations, and identifiers. Toronto has been explicitly marked in all applicable figures and captions. Figure 1a is now properly referenced in the Aircraft Flight Data section. Additionally, Fig. B1 has been clarified by combining multiple station profiles while omitting White Lake (DTX), making the differences more apparent. We have added the following text in Appendix B:

*Excluding the White Lake (DTX) station from the analysis yields a closer agreement between the control (CNTL) and the deposition-adjusted control (CNTL DA) simulations and the observational distribution. This discrepancy arises not from deficiencies in the sounding data, but from GEM's limited ability to represent the extreme ice-supersaturated conditions frequently observed over DTX.*

7. Why do you need to average over a 5 km x 5 km domain around the soundings. I thought the sounding positions are recorded (by GPS) during the radiosonde measurements versus time and, hence, known?

We use the exact location of each radiosonde to determine its position within the model domain. To account for model uncertainty in predicting ice supersaturation, we also include a 5 km $\times$ 5 km area surrounding the balloon's location. We added the following to the manuscript for clarity:

*Knowing the balloon's trajectory enables matching its observations to the nearest model grid point in both space and time. For example, a balloon launched at 12 UTC requires several minutes to reach cruising altitude, so temporal alignment is also necessary. The drift distance was therefore calculated between the launch site and the 300 hPa pressure level (approximately jet cruising altitude) for multiple stations at 12 UTC on 25 Nov 2023*

8. A caption like "3.1.2 The outlier: DTX sounding" implies that the reader already knows what a DTX sounding is. Where can I see the DTX sounding position?

The reference to the Station Identifier is in Table 1. See more in comment 6 above. We have made several new additions to help the reader.

9. Line 280:do you mean Fig 4? Line 283: do you mean Fig 5?

Yes, thank you.

10. Fig 5 is insufficiently explained. There are 8 panels which are grouped into 4 subpanels. The various panels are not explained. What do they show? The upper parts of these panels show color pots. What do the colors mean?

We have modified Fig. 5 and added the following text to the manuscript:

*On this day, these low clouds significantly attenuated the ceilometer signal at CYYZ, Toronto, making it nearly impossible to retrieve data from cirrus clouds between 1300 UTC and 1700 UTC (Fig. 4). However, a descending high cloud base was still noticeable, aligning with the descending moist layer observed in the CNTL and CNTL DA simulations (Fig. 5a and 5b, top panels). The CNTL and CNTL DA simulations show three panels each of time-height vertical profiles of $RH_i$ and the corresponding CINC for A321 (middle panels) and B747 aircraft (bottom panels).*

*The CNTL DA simulation captures the ice-supersaturated layer, favoring persistent contrail formation, between 8 km and 10 km, while the CNTL simulation shows no ice-supersaturation in this altitude range. Here, aircraft-specific differences become evident: the A321 forms contrails at $RH_i \geq 100\,\%$, whereas the heavier B747 requires $RH_i \geq 102\,\%$. In these marginally ice-supersaturated conditions, the B747's initial number of emitted ice crystals sublimates within the descending vortex. To produce contrails under the same ambient conditions ($T$, $P$, $RH_i$, atmospheric stability) observed between 1200 UTC and 1230 UTC, the B747 would require ambient temperatures $\sim 2\,^{\circ}C$ lower.*

*After 1445 UTC, the CNTL simulation remains mostly ice-subsaturated to only weakly ice-supersaturated (maximum $RH_i \approx 104\,\%$), supporting only a shallow layer with a CINC of $\sim 0.4\ cm^{-3}$ for an A321 aircraft. In contrast, the CNTL DA simulation develops a pronounced ice-supersaturated region ($RH_i$ up to 112\%) conducive to persistent contrail formation near 10 km. Under these conditions, contrails from the A321 appear first at $RH_i \geq 100\,\%$, followed by those from the B747 around 1515 UTC as $RH_i$ rises to $\approx 102\,\%$. The CNTL DA simulation produced deeper contrail forming region with enhanced CINC up to 1.4 $cm^{-3}$ for both aircraft types compared to the CNTL simulation.*

11. Fig 6 is also hard to digest. The axes are not explained. What is CINC (cm-3)? How can a reader digest headings like "Soundings for APX A321 aircraft"?

Fig. 6 was removed. CINC is now clearly defined as contrail ice number concentration in other figure captions as well as in the text.

12. I simply do not understand what you want to show. Fig 7: What is ice particle survival (in percent)? How is it computed, and why is it important?

Fig. 7 (now Fig. 6) has been modified to make it clearer. Ice particle survival has been removed because it is not important.

**References**

---

## Author Comment (AC2)

**Improving Forecasts of Persistent Contrails through Ice Deposition Adjustments**

Zane Dedekind[1], Alexei Korolev[1], and Jason A. Milbrandt[1]

[1]Meteorological Research Division, Environment and Climate Change Canada, Toronto, Ontario, Canada

**Correspondence:** Zane Dedekind (zane.dedekind@ec.gc.ca)

We sincerely thank Reviewer 2 for the constructive and very detailed feedback. The suggestions and comments considerably improved the quality of the manuscript.

We have substantially revised the preprint in response to the reviewers' comments, incorporating two main updates:

1. A full solution for calculating the deposition coefficient-based ($\alpha_D$-based) reduction factor has been implemented. $\alpha_D$ depends on temperature, pressure, humidity, and ice particle radius. These are now referred to as the deposition-adjusted (DA) simulations, replacing the previous sensitivity experiments Dep_0.6, Dep_0.8, and Dep_0.9, which are no longer required (Suggested by reviewer 2).

2. The analysis has been extended beyond a single emission index (EI = $1 \times 10^{15}$) to a range of EI values representing soot-rich to soot-poor regimes, allowing assessment of their impact on contrail evolution (Suggested by reviewer 1).

Below we present a detailed response with the reviewer comments in black, our responses in blue and additions to the manuscript in blue italics.

**Major comments**

1. From my perspective, the manuscript has two separate topics:

   (a) Better forecasting relative humidity and ice supersaturation by adjusting the deposition coefficient. As a consequence, contrail-forming areas are better predicted, which is a valuable goal.

   (b) A case study comparing contrail occurrence in model and multi-faceted observations.

   Both topics are relevant, but it should be made clearer that both topics are somewhat independent of each other. For the first topic, a less detailed description of contrails would suffice to identify ice-supersaturated regions. What's your opinion about my perspective?

   We appreciate the reviewer's perspective. While points 1a and 1b address distinct aspects of the study, they are inherently connected. Accurate forecasting of ice-supersaturated regions is essential for reliable contrail prediction. For example, in our CNTL simulations, underestimating ice supersaturation led to poor contrail representation if any at all. Thus,

although these are separate topics, improved simulation of ice supersaturation directly supports more accurate contrail forecasting. We agree that this distinction could be made clearer in the manuscript and will revise the last paragraph of the Introduction (giving the paper overview) to better delineate and connect these topics. Therefore, we modified the last paragraph in the Introduction:

*Section 3.1 focuses on the role of the deposition coefficient. Sections 3.1.1 and 3.1.2 compare the control deposition-adjusted (CNTL DA) simulation to the control (CNTL) simulation, in which the deposition coefficient is set to unity, and examine the resulting impacts on upper-tropospheric $RH_i$. Section 3.1.3 contrasts the two simulations over the Toronto region and discusses their implications for persistent contrail-forming conditions. Section 3.2 investigates the sensitivity of contrail formation and persistence to varying soot emission regimes within the DA simulations. Section 4 discusses the broader implications of these findings, followed by a summary of key results and recommendations for improving contrail representation in weather and climate models. Appendix A provides the physical basis and equations describing the depositional growth of ice crystals and the influence of the deposition coefficient $\alpha_D$ under ice-supersaturated conditions.*

We realized that the description could be more concise and have therefore already shortened it by moving the $\alpha_D$ calculations to the Appendix. In the Appendix, we provide the theoretical basis for ice crystal growth under ice-supersaturated conditions. In this revision, we also followed the reviewer's suggestion to include a full analytical solution to GEM for calculating $\alpha_D$, rather than heuristically adjusting $\alpha_D$ and comparing simulations based on heuristic parameter choices.

2. It is not clear how you define contrail persistence. It seems that contrails that have enough ice crystals at the end of vortex phase are called persistent? In GCMs or in contrail plume models, contrails are typically initialized after the vortex phase. Hence, 5-min old contrails may be considered being part of an extended formation stage. Then, persistence relates to contrails with larger lifetimes (let's say e.g. at least 60min, but choosing a precise threshold is certainly subjective). Similarly, around line 80 you state that contrail formation covers the vortex phase. This would contradict your persistence definition (tage being larger than around 5min). Please define formation and persistence and use it consistently. Check also line 399, it may be beneficial to describe in the summary section again precisely what is meant with formation and persistence. Typically, SAC only refers to temperature-controlled formation criterion and not really to persistence in ice-supersaturated regions. Please, be verbose that every reader knows exactly what is meant.

We clarified the standard definition presented in the Introduction:
*While some contrails dissipate quickly, persistent contrails form when the environmental air is ice-supersaturated, allowing ice crystals to grow rather than sublimate. Such persistent contrails may either retain their linear shape or spread into contrail cirrus, thereby contributing to cloud cover and significantly altering the Earth's radiative balance (Kärcher, 2018; Lee et al., 2023).*

Thank you for pointing this out. We had not previously provided a clear definition of persistent contrails as used in CoAT. We have now added our definition in the Contrail model: CoAT section:
*This ensures that the relative humidity over water exceeds the critical threshold required for condensation of exhaust*

*water vapor. Under these conditions, water vapor condenses and freezes onto soot and ambient aerosol particles, forming ice crystals that subsequently grow by vapor deposition. If ice supersaturation persists (i.e., $RH_i$ exceeds 100 %), contrails can remain and evolve into long-lived cirrus. Although Li et al. (2023) demonstrated that contrails may persist under ice-subsaturated conditions, for consistency with the parameterization of Unterstrasser (2016)—which assumes formation and growth under ice-supersaturated conditions—we restrict persistence to regions where ice supersaturation is maintained.*

In the Summary and Conclusions, the following was added: *The Contrail Avoidance Tool (CoAT) first applies the Schmidt–Appleman Criterion (SAC) to identify regions favorable for contrail formation and then utilizes the wake vortex model where both SAC conditions and ice supersaturation are met to diagnose persistent contrails and their properties under different soot emission regimes.*

3.  (a) You determined an optimal deposition coefficient for one synoptic scenario: How universally valid is this value? Using GEM-P3 for other synoptic scenarios, would you obtain a similar value? Is your optimal value also relevant for other microphysical models, or do you consider it to be only a tuning parameter of your P3 model?

    In response to the reviewer's concern in point 4, we have implemented a direct calculation of $\alpha_D$ within P3 rather than heuristically modulating $\alpha_D$. By diagnosing $\alpha_D$ as a function of temperature, pressure, ice particle radius, and humidity, the formulation becomes physically based and broadly applicable to other synoptic scenarios and microphysical models, rather than serving as a tuning parameter specific to P3. The description of the modifications made to the P3 cloud microphysics scheme is explained in Section 2.2.2 and Appendix A.

    (b) Please also state explicitly that your contrail initialization does not depend on the deposition coefficient, i.e. your CoAT model is independent of αd. Both, ice crystal formation and ice crystal loss during the vortex phase may, however, depend on αd but this is neglected in the parametrisation of Unterstrasser (2016). Lewellen (2014) looked at a variation of the deposition coefficient.

    We have added the following to the text:

    *The $\alpha_D$ is used only within the P3 microphysics scheme and is not included in the CoAT framework; hence, ice crystal formation and loss during the vortex phase are independent of $\alpha_D$.*

4. The deposition coefficient is varied only implicitly, see description Appendix A1. I wonder whether it wasn't technically possible to directly vary the parameter within P3. This would simplify the presentation of the results and remove the uncertainty by the non-ideal relation between deposition coefficient and mass growth rate for differently sized ice crystals (e.g. the appendix would be superfluous).

    Yes, this is correct and fair. We have implemented a direct calculation of $\alpha_D$ within P3 rather than heuristically modulating $\alpha_D$. The description can be found in Section 2.2.2. We found it necessary to retain the additional description in

Appendix A; however, it has been revised and clarified. We neglected to mention that we only applied the deposition adjustment $T < -38\,°C$. We now clarify that the reduction factor was applied only for upper tropospheric conditions, and have added this to the manuscript.

*Line 178. Thedeposition adjustment was implemented only in the upper troposphere for $T < -38\,°C$ consistent with laboratory and modeling studies (Fukuta and Takahashi, 1999; Gierens et al., 2003; Lohmann et al., 2008; Skrotzki et al., 2013; Lamb et al., 2023; Kärcher et al., 2023)*

5. I am not sure whether all relevant information is given to understand how contrails are initialized in your model. You mention Unterstrasser (2016), which provides a parametrization of ice crystal survival fraction. What's your choice of the initial ice crystal number (prior to loss)? Does this number depend on ambient conditions? (for ambient temperature close to the SAC temperature, not all soot particles are activated (Lewellen, 2020, Bier et al, 2022)). How do you determine number concentrations n, which requires the specification of a contrail cross-section? How do you determine the ice mass concentration m? Do you specify a mean mass to obtain m from n? How are contrails initialized in subsaturated air, for which Unterstrasser (2016) makes no prediction? Moreover, Unterstrasser (2016) assumes an icesupersaturated layer with a constant RHi value. As a consequence, the parametrization was not made for shallow layers where the wake vortices move out of the moist region and for which it seems you apply it. At the end of section 2.2.3 you write that the surviving ice crystals grow through deposition. Due to the very short paragraph, I am not sure whether you really simulate contrails beyond the vortex phase? Then I would assume that you had to feed the contrail properties into P3, which is however not explicitly stated. You also write that GEM advects the number and mass concentrations? Are the contrail ice crystals treated as passive tracers? Or do you really include microphysical processes? Could you please clarify this.

Our description was lacking in several ways. We have added the relevant details on the choice of the initial ice crystal number (prior to losses), which in CoAT does not depend on ambient conditions. A full description has been included in the CoAT section, and a brief summary of these additions is provided here.

The initial contrail ice number ($N_0$) and water vapor emission rate ($I_0$) were estimated from aircraft properties and emission indices, following Unterstrasser (2016). After the vortex phase, the surviving ice crystals ($CI_{surv}$) were obtained by applying a survival fraction ($f_{Ns}$). The resulting contrail ice number (CINC) and ice mass (CQI) concentrations were determined using the flight distance within each grid box and dividing by the grid volume, and then distributed along the flight path within the corresponding GEM grid cells. Sensitivity simulations were performed across different soot emission regimes (Very-High-Soot, High-Soot, Normal-Soot, Low-Soot) to assess how soot number emissions influence contrail ice formation and persistence. The sensitivities are tabled in Table 3.

Regarding the use of Unterstrasser (2016) only in shallow layers: We spread the contrail over multiple grid boxes because the simulated contrail depth can exceed the vertical extent of a single model layer in GEM. The Unterstrasser (2016) scheme assumes a constant-RHi layer, but in GEM the vertical grid spacing is often shallower than the diagnosed contrail depth. Confining the entire contrail ice content (CINC and CQI) to one grid cell would therefore overestimate

local ice concentrations. Distributing the ice content to a lower grid box (typically the layer below the flight level) provides another representation of contrail vertical extent, and accounts for dynamical effects such as shear-induced vertical dispersion. We recognize that both approaches have their limitations, but we adopted the latter approach.

Furthermore, we use CoAT only in ice-supersaturated regions to simulate persistent contrails, while contrails are not simulated beyond the vortex phase. We have removed the short paragraph at the end of section 2.2.3, which caused the confusion.

Once the survived contrail ice crystals have been determined after the wake vortex phase they are passed into the first ice category of P3, they are subjected to all microphysical processes simulated by P3. Therefore, they are not just tracers.

**Minor comments**

1. The title of the study could be refined to "Improved forecast of persistent contrail occurrence ...", because you do not simulate persistent contrails.

   We do simulate persistent contrails and have made the text clear. See point 2 in Major comments.

2. Line 52: turbulence is typically not a crucial process. Shear and sedimentation are much more important for contrail spreading, see Lewellen (2014) or Unterstrasser & Gierens (2010).

   Thank you for the information. We have added the following to the description of turbulence:
   *Nevertheless, turbulence is generally of secondary importance, with shear and sedimentation identified as the dominant processes controlling contrail spreading (Lewellen, 2014; Unterstrasser and Gierens, 2010).*

3. I do not understand why precise time periods and drift distances are provided for 300hPa.

   Our radiosonde–model comparisons focus on jet cruising altitudes. To accurately assess the simulated atmospheric state, it is important to account for where the radiosonde will be located and how long it takes to reach cruising levels. Therefore, we extract model vertical profiles at the radiosonde's position when it reaches 300 hPa. Without this procedure, the radiosonde–model comparisons would be less reliable and more prone to error. We have added the following to the text:
   *Knowing the balloon's trajectory enables matching its observations to the nearest model grid point in both space and time. For example, a balloon launched at 12 UTC requires several minutes to reach cruising altitude, so temporal alignment is also necessary. The drift distance was therefore calculated between the launch site and the 300 hPa pressure level (approximately jet cruising altitude) for multiple stations at 12 UTC on 25 Nov 2023.*

4. The definition of $G$ is introduced in Schumann (1996), not in Schumann (2012).

   We corrected the reference.

5. Line 211: you may cite Lewellen (2012) instead of Jensen (2024).

   We included Lewellen (2012)

6. Figure 2 says 5 km, but the text states 30 km. What is true? The 5 km × 5 km is a mistake. The captions in Figure 2 and 3 now says:

   *5 km × 5 km*

7. Line 317: what are the numbers in the square brackets?

   It is the range of values that are within the 5 km × 5 km. domain. We have in the meantime removed this whole section because the other reviewer suggested we include simulations of different soot emissions.

8. Is it correct that $\alpha_d$ is implicitly defined by Eqs. A3 and A4?

   Yes, $\alpha_D$ is implicitly defined in the nonlinear expressions of Eqs A3 and A4. Starting from an initial guess of $\alpha_D = 0.1$, the Newton–Raphson method is applied to obtain the numerical solution.

9. Section A1 uses radius, Figure A1 uses diameter. Moreover, the axis titles are not "nice" (long, different styles, etc.).

   We changed everything to radius for better consistency. We made Fig. A1 less "cluttered" and only show the results for $T = 225$ K and $P = 275$ hPa with a varying $RH_i$. The description in Appendix A1 was modified:
   *Figure A1 shows the results under typical cruising-altitude conditions. For example, in young cirrus clouds with particle diameters of 12–25 µm at $RH_i = 110$ %, the depositional mass growth ratios range from 0.9 to 0.8, indicating a 10–20 % reduction relative to the reference case. At $RH_i = 105$ % for the same particle sizes, the ratios decrease to 0.85–0.65, reflecting slower depositional growth.*

10. In general, I believe the font sizes in the figures are too small.

    We updated all the figures. Figure 1b and 3c have been update with appropriate legends to show where each station, including Toronto where the ceilometer is located.

11. Line 55: Unterstrasser (2016) provides a parametrization of early contrail properties. Contrail-cirrus simulations are presented in Unterstrasser et al. (2017a,b) or Lewellen (2014a,b).

    Thanks for this correction. We modified our section to reflect the suggestion.
    *Lewellen et al. (2014) and Lewellen (2014) employed large-eddy simulations with size-resolved microphysics to follow*

*contrails from their formation a few wing spans behind the aircraft through many hours of evolution, highlighting the roles of turbulence, crystal loss, and radiative feedbacks. Unterstrasser (2016) complemented this by developing a parameterization for the formation and properties of young contrails, while Unterstrasser et al. (2017a, b) extended the analysis to contrail–cirrus interactions with natural cirrus. Together, these studies using large-eddy simulations capture the progression from contrail initiation to their full development into cirrus-scale, including the conditions under which contrails lose their distinct identity once embedded in surrounding cirrus.*

12. Line 421 and 427: It depends which models you talk about. In typical LES, no phase relaxation (saturation adjustment) is used.

    That is correct and our statement was intended to reflect numerical weather prediction models. We modified the sentence to make our statement clear.

13. The parametrization of Unterstrasser (2016) has been applied in regional and global-scale models. You may find similar findings on ice crystal loss in those studies (Gruber et al., 2017; Bier & Burkhardt, 2022).

    Indeed, and thank you.

**Technical corrections**

14. A temperature cannot be warm, cooler, or colder. This is a feature of the air mass. Temperature is low/high. Similarly, a rate is not fast, only the process that is described.

    We made the corrections in the manuscript.

15. I would not refer to wake vortices as wake turbulence. Once the vortices break up and no coherent dynamical structures exist, then the elevated turbulence intensity might be called wake turbulence.

    We agree and have corrected our statements.

16. I believe you forgot to delete the sentence in lines 191 & 192.

    We removed the sentense.

17. Figure 7 should zoom into the 200 to 400 hPa layer. In the present form, most areas in the panels are just white and wasted space.

    The figure, noe Figure 6, was adapted.

18. Make sure to use the same subscript: you use $\alpha_d$ and $\alpha_D$.

    Done.

19. I prefer a mathematically correct notation in Eq. A4. One can use units in equations ($s_{\mathrm{crit}}/\% = \ldots \Delta T/\mathrm{K}$).

    Done.

20. Figure A1 should extend the y-axis down to 0.0 in all panels.

    The figure was simplified by removing panels that were not discussed, and the remaining information was de-cluttered for clarity.

21. Please check your reference section thoroughly; e.g., $CO_2$ in line 562.

    Good catch. Thank you.:

22. Line 150: Why is a refinement of the tropical troposphere relevant in your study?

    The word "tropical" should not be there and have been removed. A refinement of the troposphere is relevant because it allows for a more accurate representation of ice-supersaturated regions. At coarser vertical resolutions, thin supersaturated layers may be poorly resolved or entirely missed, whereas a finer resolution increases the likelihood of capturing these features more realistically. We modified our statement slightly.
    *...the vertical resolution in the upper troposphere with grid spacings of ~230m at 300 hPa for a more accurate representation of ice supersaturated regions.*

23. Line 390: from STUDY XX?

    We incorporated the references.

24. Line 405: for the summary section, it might be useful to explain again what $Dep_{0.8}$ refers to.

    We have removed these sensitivity simulations, but we have defined all acronyms again in the Summary section so that it is easier to follow.

25. Line 424: Should this be $0.2\,\mathrm{mm} = 200\,\mu\mathrm{m}$?

    You are correct, and we corrected this error.

**References**

Fukuta, N. and Takahashi, T.: The Growth of Atmospheric Ice Crystals: A Summary of Findings in Vertical Supercooled Cloud Tunnel Studies, https://journals.ametsoc.org/view/journals/atsc/56/12/1520-0469_1999_056_1963_tgoaic_2.0.co_2.xml, 1999.

Gierens, K. M., Monier, M., and Gayet, J.-F.: The deposition coefficient and its role for cirrus clouds, Journal of Geophysical Research: Atmospheres, 108, https://doi.org/10.1029/2001JD001558, 2003.

Kärcher, B.: Formation and radiative forcing of contrail cirrus, Nature Communications, 9, 1824, https://doi.org/10.1038/s41467-018-04068-0, 2018.

Kärcher, B., Jensen, E. J., Pokrifka, G. F., and Harrington, J. Y.: Ice Supersaturation Variability in Cirrus Clouds: Role of Vertical Wind Speeds and Deposition Coefficients, Journal of Geophysical Research: Atmospheres, 128, e2023JD039 324, https://doi.org/10.1029/2023JD039324, 2023.

Lamb, K. D., Harrington, J. Y., Clouser, B. W., Moyer, E. J., Sarkozy, L., Ebert, V., Möhler, O., and Saathoff, H.: Re-evaluating cloud chamber constraints on depositional ice growth in cirrus clouds – Part 1: Model description and sensitivity tests, Atmospheric Chemistry and Physics, 23, 6043–6064, https://doi.org/10.5194/acp-23-6043-2023, 2023.

Lee, D. S., Allen, M. R., Cumpsty, N., Owen, B., Shine, K. P., and Skowron, A.: Uncertainties in mitigating aviation non-$CO_2$ emissions for climate and air quality using hydrocarbon fuels, Environmental Science: Atmospheres, 3, 1693–1740, https://doi.org/10.1039/D3EA00091E, 2023.

Lewellen, D. C.: Analytic Solutions for Evolving Size Distributions of Spherical Crystals or Droplets Undergoing Diffusional Growth in Different Regimes, Journal of the Atmospheric Sciences, 69, 417–434, https://doi.org/10.1175/JAS-D-11-029.1, 2012.

Lewellen, D. C.: Persistent Contrails and Contrail Cirrus. Part II: Full Lifetime Behavior, Journal of the Atmospheric Sciences, 71, 4420–4438, https://doi.org/10.1175/JAS-D-13-0317.1, 2014.

Lewellen, D. C., Meza, O., and Huebsch, W. W.: Persistent Contrails and Contrail Cirrus. Part I: Large-Eddy Simulations from Inception to Demise, Journal of Atmospheric Sciences, 71, 4399–4419, https://doi.org/10.1175/JAS-D-13-0316.1, 2014.

Li, Y., Mahnke, C., Rohs, S., Bundke, U., Spelten, N., Dekoutsidis, G., Groß, S., Voigt, C., Schumann, U., Petzold, A., and Krämer, M.: Upper-tropospheric slightly ice-subsaturated regions: frequency of occurrence and statistical evidence for the appearance of contrail cirrus, Atmospheric Chemistry and Physics, 23, 2251–2271, https://doi.org/10.5194/acp-23-2251-2023, 2023.

Lohmann, U., Spichtinger, P., Jess, S., Peter, T., and Smit, H.: Cirrus cloud formation and ice supersaturated regions in a global climate model, Environmental Research Letters, 3, 045 022, https://doi.org/10.1088/1748-9326/3/4/045022, 2008.

Skrotzki, J., Connolly, P., Schnaiter, M., Saathoff, H., Möhler, O., Wagner, R., Niemand, M., Ebert, V., and Leisner, T.: The accommodation coefficient of water molecules on ice – cirrus cloud studies at the AIDA simulation chamber, Atmospheric Chemistry and Physics, 13, 4451–4466, https://doi.org/10.5194/acp-13-4451-2013, 2013.

Unterstrasser, S.: Properties of young contrails; a parametrisation based on large-eddy simulations, Atmospheric Chemistry and Physics, 16, 2059–2082, https://doi.org/10.5194/acp-16-2059-2016, 2016.

Unterstrasser, S. and Gierens, K.: Numerical simulations of contrail-to-cirrus transition – Part 1: An extensive parametric study, Atmospheric Chemistry and Physics, 10, 2017–2036, https://doi.org/10.5194/acp-10-2017-2010, 2010.

Unterstrasser, S., Gierens, K., Sölch, I., and Lainer, M.: Numerical simulations of homogeneously nucleated natural cirrus and contrail-cirrus. Part 1: How different are they?, Meteorologische Zeitschrift, pp. 621–642, https://doi.org/10.1127/metz/2016/0777, 2017a.

Unterstrasser, S., Gierens, K., Sölch, I., and Wirth, M.: Numerical simulations of homogeneously nucleated natural cirrus and contrail-cirrus. Part 2: Interaction on local scale, Meteorologische Zeitschrift, pp. 643–661, https://doi.org/10.1127/metz/2016/0780, publisher: Schweizerbart'sche Verlagsbuchhandlung, 2017b.